

# Rapid regional assessment of rock glacier activity based on DInSAR wrapped phase signal

Federico Agliardi[1], Chiara Crippa[1,2], Daniele Codara[1], Federico Franzosi[1]

1 Department of Earth and Environmental Sciences, University of Milano-Bicocca, Milano, 20126, Italy
2 Institute of Earth Observation, EURAC Research, Bolzano, 39100, Italy

*Correspondence to*: Federico Agliardi (federico.agliardi@unimib.it)

**Abstract.** Alpine periglacial landforms like rock glaciers and protalus ramparts are key indicators of the state of permafrost occurrence and its climatic implications. These landforms are characterized by complex deformation mechanisms and temporal trends, that can evolve towards destabilization. A quantitative evaluation of their activity is thus fundamental in climatological and geohazard perspectives. Spaceborne interferometric synthetic-aperture radar (InSAR) techniques have provided powerful tools to document the surface deformations of periglacial features, yet their rapid and reliable application over large areas is still limited.

We propose a novel, semi-automated methodology that combines wrapped phase deformation signals obtained from differential interferometric synthetic-aperture radar (DInSAR), available information on permafrost extent, geomorphological data and multivariate statistics to characterize the activity of 514 periglacial landforms over approximately 1000 km² in Upper Valtellina (Italian Central Alps). We process Sentinel-1 A/B SAR images with increasing temporal baselines (12 to 120 days) to generate 124 interferograms in ascending and descending geometries. We analyse the statistical distribution of wrapped interferometric phase to assess the state of activity of each periglacial landform through an objective Activity Index. This is combined with regional-scale information on permafrost occurrence to classify periglacial landforms based on their activity on different temporal scales. We define four activity classes, validated with field geomorphological observations, and related to their environmental controls through multivariate statistical analysis. Our results demonstrate the potential of using wrapped SAR interferometric phase to rapidly update periglacial landform inventories and track the evolution of the alpine cryosphere.

## 1 Introduction

Ongoing climate change strongly impacts the alpine cryosphere, with major implications for mountain landscape, hydrology, availability of water resources, infrastructure stability and durability, and geohazards (Haeberli, 2013; Beniston et al., 2018; Kääb et al., 2021). Permafrost degradation caused by climate warming changes the rheology and stability of ice-bearing soils, affecting alpine slope dynamics, sediment transport and possible destabilization (Buchelt et al., 2023). Moreover, rising ground temperature and the subsequent ice loss can reduce the water storage potential of permafrost grounds (Azocar and Brenning,



2010) and increase subsurface water triggering fast shallow slope instabilities and contribute to slope-scale natural hazards (Stoffel and Huggel, 2012; Scapozza et al., 2014; Bodin et al., 2016; Kummert et al., 2017). In this context, the study of periglacial landforms like rock glaciers and protalus ramparts is of key importance, as their distribution and activity provide critical indicators of the past and present condition of mountain permafrost, and changes in their dynamics provide proxies of

climate change trends (Cicoira et al., 2021).

Rock glaciers and protalus ramparts are bodies of frozen debris and ice that move under gravity load in permafrost areas (Barsch, 1996), typically associated with permafrost bodies 10 to 100 meter thick (Mühll and Haeberli, 1990; Humlum, 1997). Their development requires favourable combinations of climatic and topographic conditions, precipitation regimes, and supply of debris with suitable grain size (Kääb et al., 2004). These landforms are more resilient to climate warming than glaciers,

thanks to a surface debris cover that insulates an ice rich core (Jones et al., 2019, Buchelt et al., 2023).

Rock glaciers are characterized by elongated shapes and steep fronts and, depending on the occurrence or lack of internal ice, are classified as intact or relict (Barsch, 1996). Intact rock glaciers normally have a swollen appearance, with well-developed lobes of coarse debris as well as transverse ridges and furrows, that result from compression and extension to different flow velocities and internal deformation patterns. Relict rock glaciers, that entirely lost their frozen core, tend to have flatter surface

morphology, with vegetated relic structures testifying past movements (Scotti et al., 2013, Bertone et al., 2022).

Protalus ramparts are ridges or debris ramps, whose origin is still debated, but usually related to the deformation of permafrost-bearing debris at the base of steep talus slopes hosting perennial or semi-perennial snow beds below rocky cliffs in alpine environments (Shakesby, 2004; Hedding, 2011; Scapozza et al., 2011; Scotti et al., 2013).

The movement of rock glaciers and protalus ramparts is dominated by internal permafrost creep and basal frictional sliding

(Scapozza et al., 2011; Cicoira et al., 2021), resulting in spatially heterogeneous and extremely variable surface displacement rates. Typical rates of movements of rock glaciers and protalus ramparts usually range between few centimetres and several decimetres per year, exhibit seasonal variations, and can reach some meters per year in case of ice-rich landforms on steep slopes (Haeberli, 1985; Haeberli et al., 2006; Scapozza et al., 2011). Thus, a complete characterization of the state of activity of periglacial landforms requires a quantitative analysis, conducted over different temporal scales and able to reflect the

contributions of the different underlying deformation mechanisms.

Although a quantitative evaluation of displacement rates is a key component of the study of creeping periglacial features, a proper *in situ* assessment of their state of activity remains challenging, due to their difficult site accessibility, geomorphological and dynamic complexity. These factors limit the possibility to conduct geophysical surveys, boreholes and ground-based displacement measurements, that remain confined to few case studies (Bearzot et al., 2022; Bertone et al., 2023). Regional

inventories, based on geomorphological criteria, traditionally include activity attributes of mapped landforms that rely on their surface evidence, and lack a quantitative assessment of associated spatial-temporal kinematic patterns (Scotti et al., 2013, Buchelt et al. 2023, Bertone et al., 2022), despite the major improvements in the standardization of geomorphological mapping and activity classification (RGIK, 2021-2022).





To fill this gap, remote sensing techniques, including spaceborne interferometric synthetic-aperture radar (InSAR) and offset tracking applied to optical images, have been used to retrieve displacement fields and times series for mapped periglacial features (Strozzi et al., 2020; Kääb et al., 2021; Zhang et al., 2021). On the site-specific scale, the integrated use of interferometric and offset tracking techniques allows following rock glacier activity across multiple temporal scales, from cm/yr to m/yr, and up to destabilization (Cicoira et al., 2021). On the regional scale, the International Permafrost Association (IPA) Action Group (Strozzi et al., 2020; RGIK, 2021,2022; Bertone et al., 2022) has proposed guidelines to improve rock glacier inventories in mountain regions, including a kinematic attribute based on multi-annual velocity ranges derived by differential interferometric synthetic-aperture radar (DInSAR) and local-scale monitoring data. All these approaches strongly improved the state of the art, allowing to effectively capture the displacement rates and styles of activity of periglacial features. However, they rely on the manual analysis of multiple DInSAR interferograms and satellite optical images (Kääb et al., 2021, Rouyet et al., 2021, Zhang et al., 2021, Jones et al., 2023) or the site-specific analysis of displacement time series. Thus, they are time consuming, partly subjective, and difficult to apply systematically to regional inventory datasets, that include hundreds or thousands of phenomena, especially if the analysis is updated regularly to track the progress of climate change or geohazards. We propose a novel, semi-automatic methodology that combines Sentinel 1 DInSAR wrapped phase deformation signals, available information on permafrost extent, geomorphological data and multivariate statistics to: a) characterize the state of activity of 514 rock glaciers and protalus ramparts, retrieved from a published inventory (Scotti et al., 2013) over 1000 km$^2$ in Upper Valtellina (Italian Central Alps), and: b) demonstrate the potential of raw SAR interferometric data for a fast, regional scale characterization of periglacial landform activity.

## 2 Data and Methods

### 2.1 Periglacial landform inventory

The study area includes the upper alpine sector of Valtellina (~1000 km$^2$) in the Lombardia region (Northern Italy). From a geological point of view, the area is characterised by the outcrop of metamorphic, metasedimentary, and sedimentary rocks, belonging to the Variscan crystalline basement of the former Adria passive margin and the related Permo-Mesozoic sedimentary successions, stacked in the Austroalpine structural units during the Cretaceous (Froitzheim et al., 1994).

The area is characterized by elevation range between 700 m a.s.l. and 3850 m a.s.l., high local relief and a relatively steep topography (mean slope value of 30-35°) strongly imprinted by glaciations over the Quaternary, including Last Glacial Maximum and later pulses (Ivy Ochs et al, 2008, Bini et al., 2009) and until recent times. Although rapidly shrinking, glaciers still survive at the highest elevations (Fig.1a), covering about 3% of the study area (data from Regione Lombardia, 2016).

The climate has a continental character, with average annual precipitations lower than 1000 mm and seasonal maxima in July-August and October-November (Scotti et al., 2013; Ceriani and Carelli 2000, Casale et al., 2022). Mean air temperature strongly depend on altitude, but is generally lowest in January (mean air temperature: -10 - 0°C) and highest in July (mean air temperature between: 7-15 °C) over the entire area (Casale et al., 2022).







**Figure 1:** distribution and geomorphological classification of periglacial features mapped in the study area (after Scotti et al., 2013). a) inventory map (polygons); b) simplified sketch of an intact rock glacier; c-g) examples of mapped periglacial features, locations shown in a). Imagery: Google, ©2021 Maxar Technologies.





Using aerial optical imagery, a 5m DEM and field surveys, Scotti et al. (2013) compiled a geomorphological rock glacier inventory for the entire Lombardia. Within our study area, the inventory includes 514 protalus ramparts and rock glaciers, the latter classified according to both genetic and morphological attributes (Fig.1, supplementary Figs. S1-S3). The genetic

attribute indicates the origin of the material supplied to the landform: talus rock glaciers are fed by ice-free slope debris originated by adjacent rock walls (including rockfall and rockslide deposits), while debris rock glaciers mainly rework glacial deposits (Barsch, 1996). The morphological attribute classifies rock glaciers, according to their plan shape and geometry, as lobate (length/width <1) or tongue-shaped (length/width >1; Barsch, 1996; Nyenhuis et al., 2005).

**Table 1.** Inventory of periglacial landforms, subset of Scotti et al (2013)

| Geomorphological classes | Intact | Relict | Total |
|:---:|:---:|:---:|:---:|
| Talus-tongue | 85 | 43 | 128 |
| Talus-lobate | 155 | 103 | 258 |
| Debris-lobate | 17 | 4 | 21 |
| Debris-tongue | 50 | 8 | 58 |
| Protalus ramparts | 19 | 30 | 49 |
| Total | 326 | 188 | 514 |

Scotti et al (2013) also assigned to each mapped periglacial landform an attribute of "activity" (Fig. 2a; Table 1), essentially related to the geomorphological evidence of expected inner presence or lack of permafrost ice. Intact rock glaciers are characterized by steep fronts and side slopes and furrow-and-ridge morphologies usually associated to ice occurrence (Fig.

2b,c). Intact landforms may include "active" (i.e. moving) or "inactive", i.e. in an advanced stage of permafrost degradation or lacking material supply and showing little to no movement (Barsch, 1996; Scotti et al 2013). Here we adopt the recent proposal of RGIK (2020, 2022) to replace the term "inactive" with "transitional", since landforms with these geomorphological characteristics can still move at rates lower than 10 cm/yr (Lambiel et al., 2023). On the opposite, "relict" landforms show a subdued topography, collapsed structures and growing vegetation (Fig. 2d), testifying the complete disappearance of

permafrost (Scotti et al 2013). The activity attributes by Scotti et al (2013) are based on purely geomorphological criteria and may be affected by operator bias (Brardinoni et al., 2019).

The resulting classification includes five classes of landforms (Table 1), distributed over the entire study area at altitudes between 2000 and 3500 m.a.s.l.. Relict features tend to be found at altitudes in the range 2350-2550 m (supplementary Fig. S1) and characterized by variable slope values depending on the considered class, but most frequently in the range 16-25° (IQ

range for the talus-lobate class; Fig. S2). Active landforms mostly occur between 2600 and 2800 m a.s.l. (Fig. S1) and are associated with slightly steeper slope values (20-28° IQ range for talus-lobate features; Fig. S2). Mapped landforms are characterized by variable topographic aspect (Fig.S3), with debris rock glaciers mainly facing to the N and talus-related



landforms showing a more variable aspect related to the local cliff topography. Although not been updated after 2013, the inventory provides a robust reference to identify the location and general characteristics of periglacial landforms in the area.


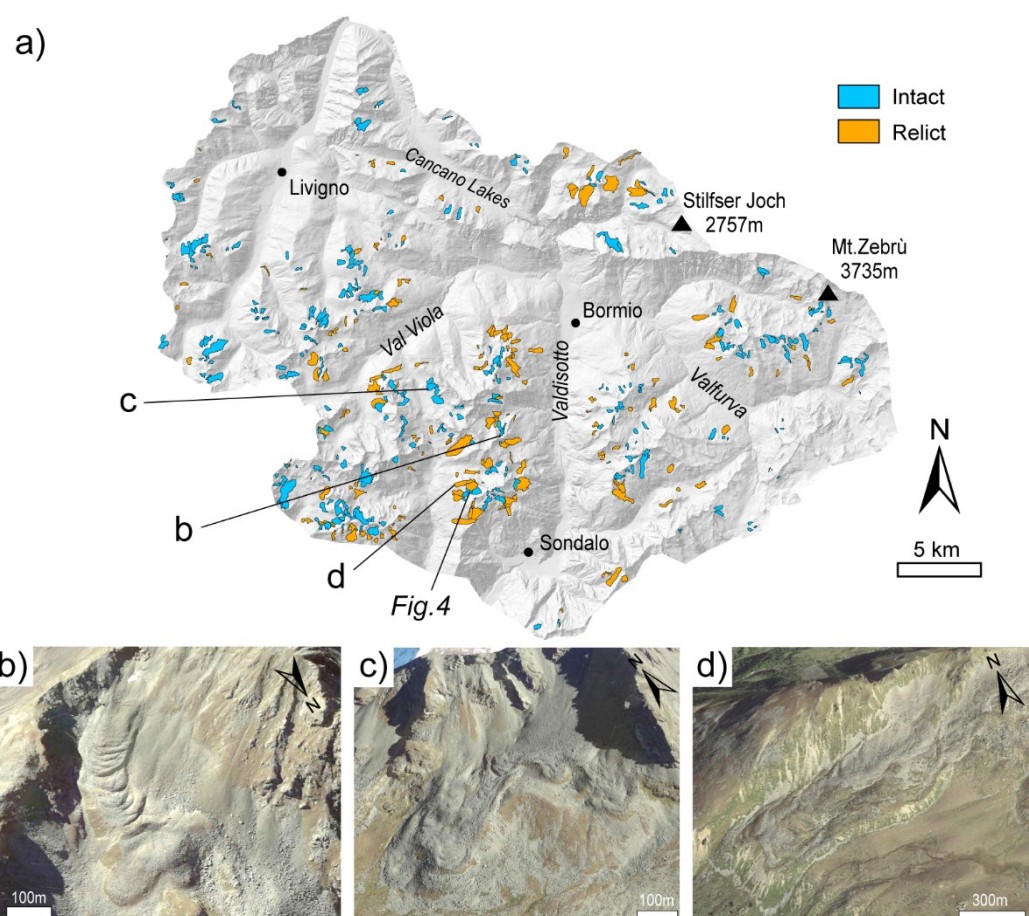

**Figure 2.** Geomorphological classification of periglacial landform activity. a) distribution of activity classes according to Scotti et al., 2013; b,c) intact landforms still host permafrost; d) relict landforms do not have a permafrost core anymore. Landform locations are shown in a). Imagery: Google, ©2021 Maxar Technologies.


## 2.2 Permafrost extent

We accounted for the likely extent of permafrost in the study area using the Alpine Permafrost Index Map (APIM; Boeckli et al., 2012). This is the result of a statistical model accounting for a set of permafrost occurrence predictors, including the mean annual air temperature, the potential incoming solar radiation, and the mean annual total precipitation (Boeckli et al., 2012).

Using an RGB colour code, the map portrays the likelihood of permafrost occurrence with respect to the local conditions (Fig.3a,b). Since permafrost conditions may have changed since the product publication, to obtain a conservative estimate of



permafrost extent we recoded the map by the red (R) band values of the APIM RGB colour code (range: 0-255), and then filtered the areas with R values below a specified threshold, to keep only the API class "permafrost in nearly all conditions". We used a threshold of R=240, calibrated by comparing the relative frequency of intact and relict features mapped by Scotti

et al. (2013) with the modelled presence or absence of permafrost (Fig.3c).

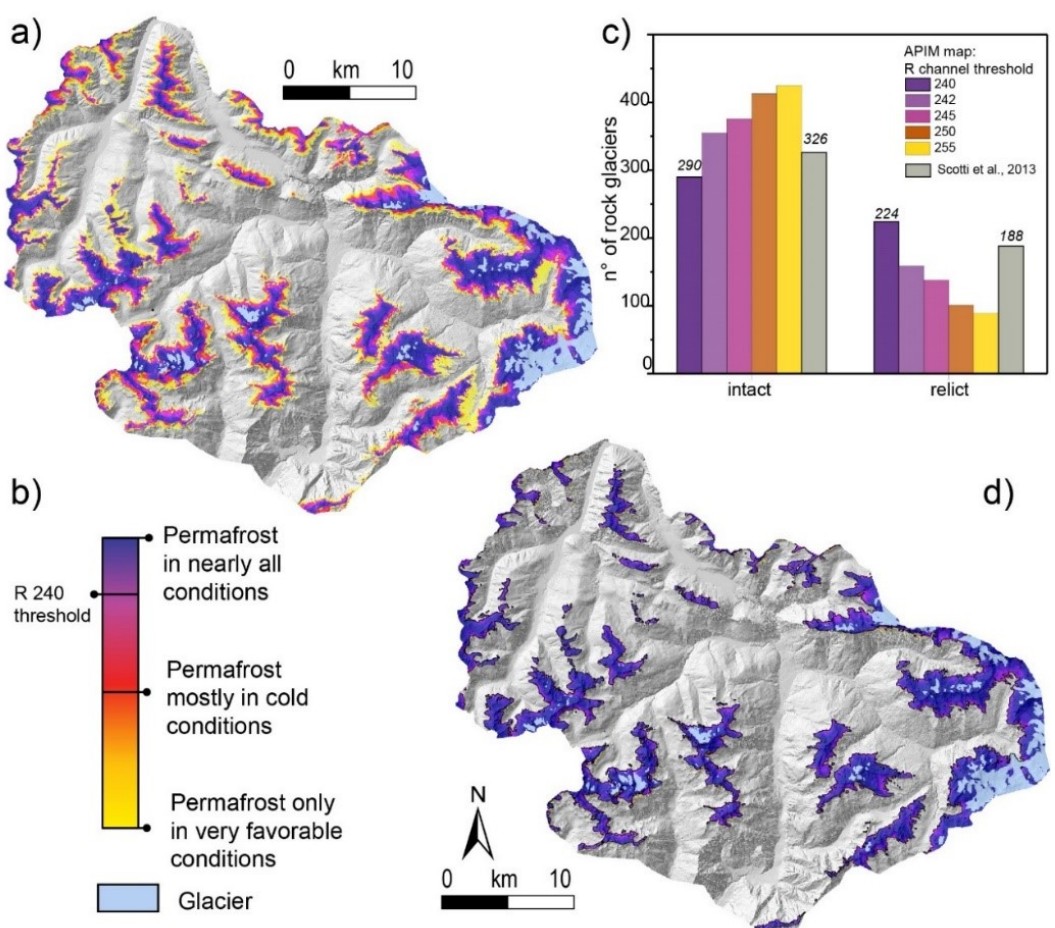

**Figure 3.** Likely permafrost distribution in the study area. for the study area. a) original Alpine Permafrost Index Map (APIM; Boeckli et al., 2012); b) likelihood classes of permafrost occurrence based on the APIM RGB colour code; c) histogram reporting the frequency of periglacial landforms with or without permafrost as a function of different threshold values of the R (red channel) value, compared with the
"intact" and "relict" classes of Scotti et al (2013); d) permafrost extent considering a R threshold of 240.

## 2.3 Surface deformations: DInSAR

We retrieved 194 Sentinel-1 A/B Single-Look Complex (SLC) radar images (C-band, wavelength: 5.56 cm), acquired in TOPS Interferometric Wide swath (IW) mode between June 2017 and October 2020, in both ascending and descending geometries



(supplementary Tables S1 and S2). DInSAR processing was then carried out using the ESA SNAP (Sentinel Application

Platform) software v.8, using the SRTM 1Sec HGT DEM for co-registration, topographic phase removal and terrain correction.

For each mapped feature, we selected the best acquisition geometry depending on the C index by Notti et al. (2014), that is a

function of topographic (slope and aspect) and satellite orbit parameters (incidence LOS angle, Θ, and orbital azimuth angle,

δ). The C index (supplementary Fig. S4) provides a measure of the fraction of movement that can be recorded by a SAR sensor,

allowing selecting the most appropriate SAR acquisition geometry to be used for each periglacial feature.

We processed 100 radar images in ascending geometry and 94 in the descending one, covering the summer-autumn periods

(June to October) of the considered years, in agreement with the guidelines of the IPA Action Group (RGIK, 2021).

Since periglacial features can move at different rates, their deformation signal can be masked by noise or lost by phase aliasing

when they move too fast (Lambiel et al., 2023). Therefore, to explore the temporal baselines of interferogram generation (Bt)

most suitable to capture the displacement rates typical of the different rock glacier deformation processes (basal sliding,

permafrost creep, active layer destabilization; Cicoira, 2021), we first generated interferograms at 6, 12, 18, 24 and 90 days,

as well as 120/140 days ("seasonal" interferograms) for year 2017 (see supplementary Tables S2 and S3).

This preliminary investigation, in agreement with other published studies (RGIK, 2021; Strozzi et al., 2020; Brencher, et al.,

2021), allowed selecting temporal baselines of 12, 24, 90 and 120/140 days (i.e. interferograms covering the maximum

observation window in the snow free period from June to October). Interferograms spanning 12-24 days show high coherence

(mean values on the features of interest >0.4) and provide signals representing short-term deformation patterns of mapped

periglacial features. On opposite, 120/140-day interferograms provide information on overall seasonal deformations.

Then, we generated 233 interferograms from images in ascending geometry and 274 from descending ones, using different

temporal baselines (Bt). We manually inspected all the products, and discarded those affected by significant decorrelation,

atmospheric phase screening (APS) and snow-cover disturbances. After this process, the number of usable products greatly

decreased. For subsequent analyses we exploited 124 interferograms (45 ascending and 79 descending), i.e. 25% of the full

set of interferograms, pointing out the limitations to DInSAR caused by the alpine environment and climatic conditions.

To improve the signal-to-noise ratio, all the interferograms were multi-looked by factors of 4 in the range and 1 in the azimuth

direction and filtered using a Goldstein filter (Goldstein and Werner, 1998).

Despite the careful image selection and processing, Sentinel-1 interferograms rarely display clear interferometric fringe

patterns on the features of interest. Their unwrapping and conversion to displacements (Costantini, 1998) is usually unfeasible

or excessively uncertain, due to abrupt phase jumps, disconnected coherent patches, etc. Therefore, we didn't apply the

unwrapping procedure over the AOI, but focused on the wrapped phase information that can be detected and mapped.

Without unwrapping, a precise measurement of displacement rates remains elusive. However, for a given temporal baseline

(Bt) and radar wavelength (λ), ranges of (constant) displacement rates can be estimated assuming that displacement between

two acquisitions are lower than λ/2 (to avoid phase ambiguity) or λ/4 (to avoid both phase and direction ambiguities; Colesanti

and Wasowski, 2006; Manconi, 2021).



## 2.4 Analysis of DInSAR wrapped phase signals

To exploit the information contained in the DInSAR wrapped phase signals, we prepared a semi-automatic procedure that
analyses phase value distributions to infer the state of activity of each mapped periglacial feature. The procedure includes four steps (Fig. 4): a) identification of stable reference areas ("rims") around each inventoried periglacial feature (Fig. 4a); b) correction of DInSAR wrapped phase values within each feature with reference to its stable reference area (Fig. 4b); c) stacking of the corrected wrapped interferograms generated with same temporal baseline Bt and median stacked phase computation, allowing mitigating atmospheric disturbances (Fig. 4c); d) analysis of the distribution of median stacked phase values inside
each rock glacier and stable area, and calculation of an Activity Index (Fig. 4d).

Based on the mapping criteria used by Scotti et al. (2013), we assumed that movements related to periglacial processes are confined within polygon boundaries, while surrounding areas, lacking evidence of permafrost deformation, are considered stable. We generated 30 m wide buffers around each mapped landform to identify "stable rims" with no visible displacement. In the case of adjacent forms or multiple rock glaciers merging into one body, the rims were cut to avoid overlapping among
different features. The narrow buffer width allowed to avoid including excessive areas of proximal talus slopes or rock walls, that would introduce noise or artefacts in the phase value. However, rim width and shape may be calibrated for specific needs.

To allow a consistent comparison among different interferograms, we corrected the DInSAR wrapped phase values ($\phi_{Li}$) inside each landform (Li) with respect to their stable rims (Fig. 4a), by subtracting the modal value of wrapped phase difference within the stable rim, $Mo(\phi_{Ri})$, from each phase value within the landform (Fig.4b). Other authors used a similar approach,
considering local stable reference points near ridges and bedrock outcrops (Brencher et al., 2020). However, they performed manual selections, hampering an automated analysis on the regional scale (i.e. for hundreds of periglacial landforms).

After this correction, all the interferograms generated with the same temporal baseline (Bt) and acquisition geometry (ascending or descending) were stacked to compute the median wrapped phase value of each pixel within each periglacial feature, $X_{Li}$ (Fig. 4c). Then, the distributions of the median stacked phase inside the landforms, $f(X_{Li})$, were compared to the
ones inside the corresponding stable rims, $f(X_{Ri})$ (Fig. 4d). We assume that values of $f(X_{Li})$ falling outside ±1 standard deviation of $f(X_{Ri})$ correspond to active deformation and, for each landform *i*, define the Activity Index AI (Equation 1):

$$AI_i = \frac{\int_{-\infty}^{\infty} f(X_{Li})dX_{Li} - \int_{-SD(X_{Ri})}^{+SD(X_{Ri})} f(X_{Li})dX_{Li}}{\int_{-\infty}^{\infty} f(X_{Li})dX_{Li}}$$
Eq.1

where:

$X_{Li}$:  median stacked wrapped phase within the periglacial landform *i*

$X_{Ri}$:  median stacked wrapped phase within the stable rim surrounding the landform *i*

$f(X_{Li})$:  probability density distribution of $X_{Li}$

$SD(X_{Ri})$:  sample standard deviation of $X_{Ri}$





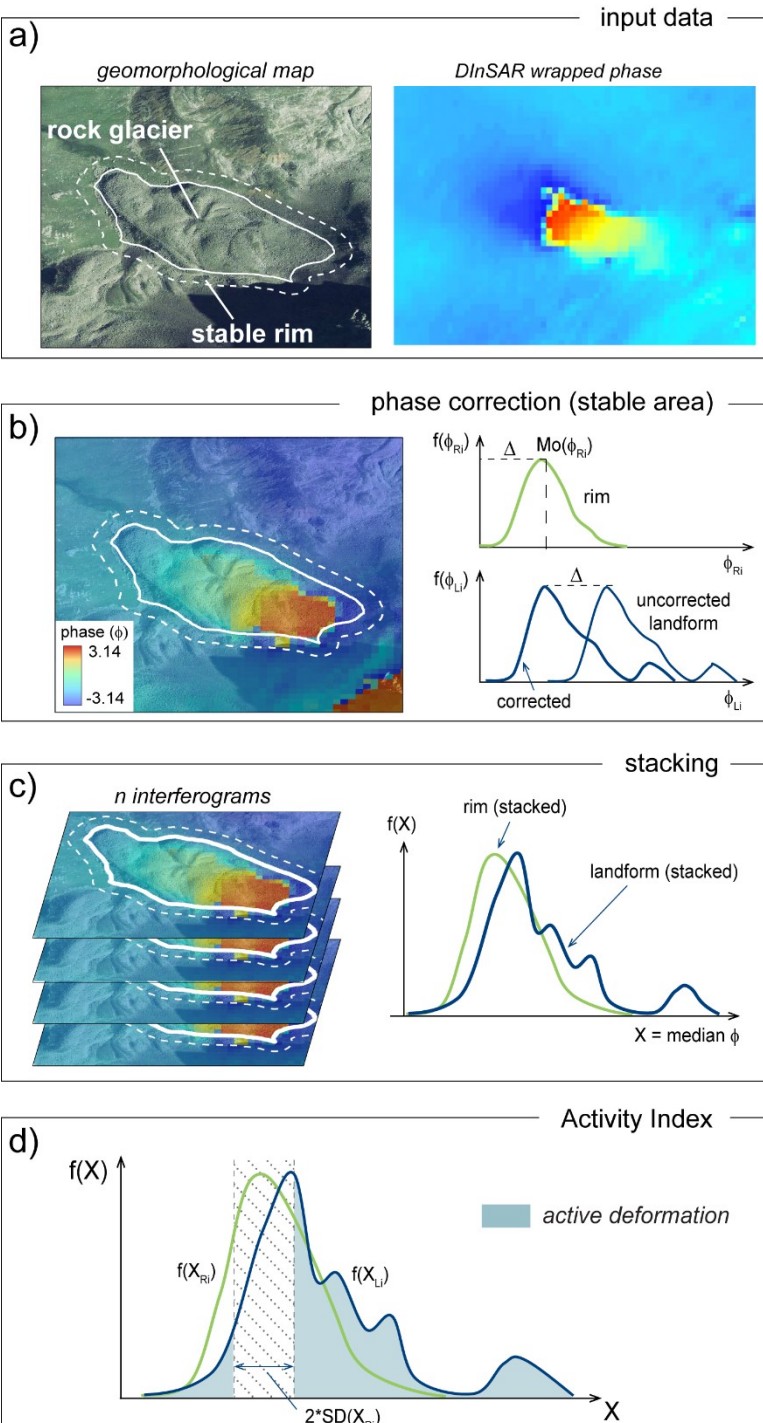

**Figure 4.** Activity Index analysis workflow. a) polygons of rock glaciers and their stable rims, and related DInSAR wrapped phase signals computed at a given temporal baseline Bt; b) landform phase correction with respect to the modal phase value of the corresponding stable rim; c) stacking (median phase values) of all the interferograms generated at the same temporal baseline; d) calculation of the Activity Index (see text for explanation). The location of the example rock glacier is outlined in Fig. 2a. Imagery: Google, ©2021 Maxar Technologies.




## 2.5 Land surface temperature (LST)

The inner part of intact rock glaciers hosts permafrost ice, which is insulated by the surficial debris cover. The active layer of rock glaciers plays a key role in regulating heat exchange with permafrost, influencing land surface temperature (LST), that is cooler during warm months in active rock glaciers with respect to transitional or relict ones (Alcott, 2020). Thus, LST is thus a useful indicator of the physical state of periglacial features, supporting the validation of their activity classifications. In this perspective we computed the LST of the areas of interest using Landsat 8 images (Table 2).

Analyses were carried out on Google Earth Engine™ (GEE) platform using the code proposed by Ermida et al. (2020) to process thermal infrared (TIR) band signals provided by Landsat 8 over the period 2013 to 2020.

**Table 2:** dataset used to compute LST in GEE. A cloud filter was added to exclude images with a cloud cover over 20%.

| Satellite | Bands | Wavelength (µm) | Dataset | ground resolution | time period |
|---|---|---|---|---|---|
| Landsat 8 (OLI; TIRS) | Red: B4 | 0.64–0.67 | C01/T1_SR | 30 m | |
| | NIR: B5 | 0.85–0.88 | C01/T1_SR | 30 m | 2013-2020 |
| | TIR: B10 | 10.6–11.19 | C01/T1_TOA | 100 m * | |

* gridded to 30 m

Mapped periglacial features are distributed across a wide range of altitudes, and several (especially those classified as relict) are covered by shrubs. As consequence, we quantified LST using an algorithm able to correct land emissivity for the surface vegetation contribution using the NDVI (Normalized Difference Vegetation Index; Malakar et al., 2018, Parastatidis et al., 2017, Ermida et al., 2020). We also used a filter to exclude all the images with cloud coverage exceeding 20% of the scene. We retrieved 53 images from GEE and processed them in Matlab™ to extract temperature statistics for selected time intervals. We computed a mean LST map for summer (June-October) and winter periods (October-June) (Fig.5 a,b), as well as an overall total mean LST (Fig 5c) of the 53 images, with no seasonal distinction. However, retrieving meaningful information for winter periods remains challenging because of the extent of snow cover in the study area.

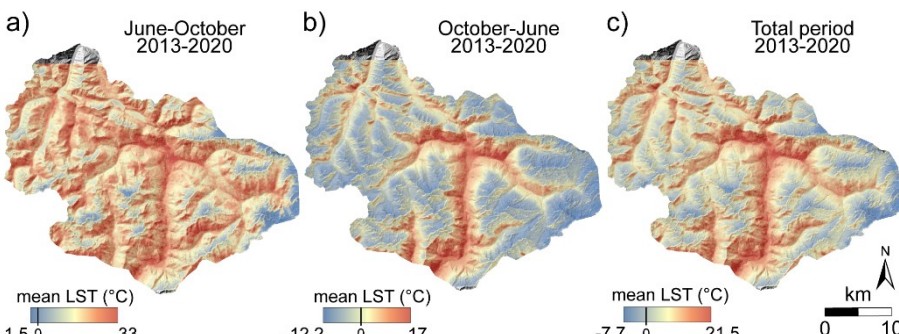

**Figure 5.** LST computed over the period 2013-2020 for the study area. a) mean LST for summer months (June-October). Negative values commonly indicate glaciers and snowfields; b) mean LST for winter periods (October-June). c) overall mean LST (no seasonal distinction).



## 2.6 Principal Component Analysis

Rock glacier dynamic and activity depend on a combination of morpho-climatic factors related to latitude, elevation, slope, aspect, as well as the amount of sediment supply from contributing sources. We used Principal Component Analysis (PCA, Hotelling 1933) to explore the relationships between these factors (Table 3) and the periglacial landform activity on different temporal scales, assessed through our methodology, as a value added to the inventory. PCA allows to explore and interpret a set of data by reducing the dimensionality of variables, i.e. by finding new variables that are linearly related to the original

ones, maximize their variance, and are uncorrelated to each other (Ballabio, 2015).

In our analysis, the number of PCs was determined according to their eigenvalue, representing the amount of variance explained by a given principal component. We considered all the PCs with eigenvalue > 1, a value indicating that the PC accounts for more variance than accounted by one of the input variables.

**Table 3:** input parameters of the multivariate statistical analysis.

| Morphometry | Label | Description |
|---|---|---|
| elongation ratio | L/W | form factor (elliptical shape with axes L and W) |
| mean elevation | mean elevation | elevation of the landform polygon centroid |
| mean aspect | Aspect | mean aspect calculated as circular mean of each pixel |
| area | Area | area of the periglacial landform polygon |
| slope | Slope | mean slope in each polygon |
| **Remote sensing** | **Label** | **Description** |
| Summer LST | LST Jun-Oct | Land surface temperature in the June-October period (mean 2013-2020) |
| Winter LST | LST Oct-Jun | Land surface temperature in October-June period (mean 2013-2020) |
| Total LST | LST total | Land surface temperature (overall mean 2013 to 2020) |

## 3 Results

### 3.1 Periglacial landform activity

The wide-area application of our methodology allowed us to compute an Activity Index based on the kinematic information contained in raw interferometric products (i.e. DInSAR wrapped phase maps), generated for different reference timescales

(e.g. temporal baseline, Bt). Our Activity Index (Eq. 1) is easily represented in maps (Fig 6a) and provides a non-dimensional description of landform activity, depending both on landform displacement rates and the extent of moving areas.

Nevertheless, the activity of individual periglacial features in geomorphological inventories is assessed in a classified form, requiring problem-specific classification criteria and thresholds. Here we combined two criteria, namely a threshold value ($AI_T$) of the Activity Index and the occurrence of permafrost, to reclassify the datasets into four landform activity classes



(Table 4; Fig. 6b). Three of them (active, transitional, relict) correspond to the activity classes usually adopted in rock glacier inventories (RGIK 2020, 2022), while the "moving debris" was introduced here to account for landforms that show significant displacements without hosting permafrost.

**Table 4:** activity classes resulting from the DInSAR analysis according to unique combinations of degree of activity and permafrost
occurrence. $AI_T$: Activity Index threshold

| Activity classes | degree of activity | permafrost occurrence |
|---|---|---|
| **Active** | $AI > AI_T$ | Yes |
| **Transitional** | $AI < AI_T$ | Yes |
| **Relict** | $AI < AI_T$ | No |
| **Moving debris** | $AI > AI_T$ | No |

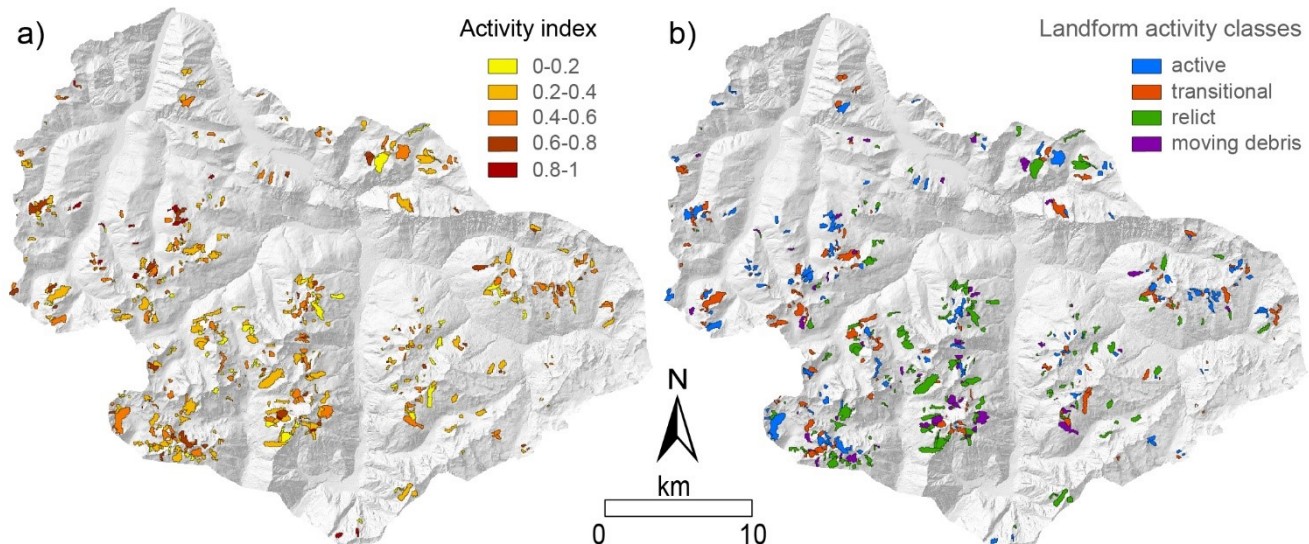

**Figure 6.** Maps of Activity Index and landform activity classes on a temporal baseline of 24 days; a) polygons classified according to the
computed values of Activity Index (0-1); b) reclassified map obtained considering an AI threshold at 0.4 and the presence of permafrost.

The final classification (Fig. 6a,b, Table 4) is sensitive to the selected value of the threshold $AI_T$ and to the temporal baseline Bt of the reference interferograms. Such sensitivity is displayed in the form of curves (Fig.7) showing the absolute number of landforms in each class of activity as a function of $AI_T$. The curves related to "active" and "transitional" landforms (Figs.7a,b)
are symmetric due to their intrinsic definition (i.e. presence of permafrost, complementary ranges of AI). The sensitivity of classification to $AI_T$ (i.e. curve steepness) is higher for values lower than 0.6, and strongly depends on Bt (i.e. 90 days, seasonal). This results in different classification results for different temporal baselines, in agreement with the different



commonly observed timescales of landform activity. The maximum separation between curves obtained with different temporal is observed for $AI_T$ values in the range 0.35-0.5. On the other hand, the curves related to the "relict" and "moving
debris" classes (Figs. 7c,d), characterized by absence of permafrost absence of permafrost, are almost completely insensitive to Bt. The final classification portrayed in the map in Fig. 6b corresponds to a value of $AI_T$ of 0.4, that allows to maximize the difference between classifications obtained for different reference timescales of process occurrence, and thus classes of displacement rate.

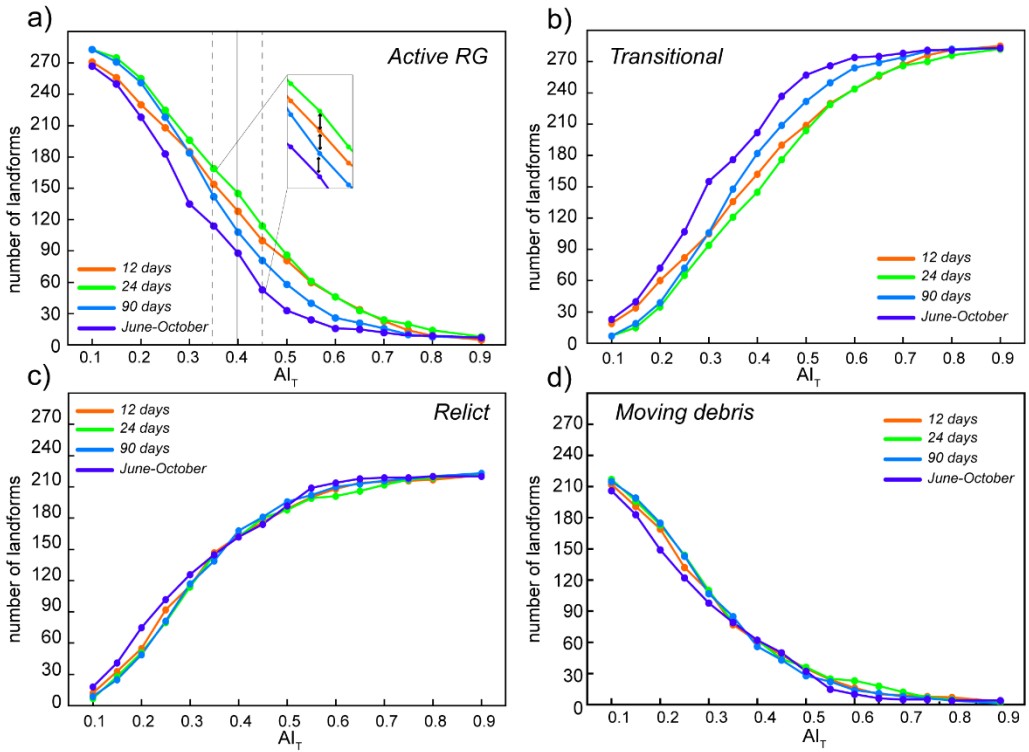

**Figure 7.** Dependence of the abundance of periglacial features in different activity classes on the threshold value of the Activity Index ($AI_T$) and the temporal baseline (Bt). a) the number of "active" and b) "transitional" features are complementary and strongly dependent on Bt, especially for $AI_T$ values in the range 0.35-0.5. c) the numbers of "relict" and d) "moving debris" features are almost independent on each other and on the temporal baseline Bt.


The strong sensitivity of the number of landforms classified as "intact" (i.e. active and transitional) on Bt suggests that their activity is related to underlying mechanisms occurring at different rates, better captured over few days (active layer movement, basal frictional sliding) or longer periods (internal permafrost creep). On opposite, the independence of the "moving debris" class on Bt may suggest that the activity of related features is not directly linked to permafrost but controlled by the frictional
properties of their granular mixture.





### 3.2 Styles of activity and comparison with geomorphological evidence

The number of periglacial landforms assigned to each activity class by our methodology is reported in Table 5. The populations of "active" and "transitional" classes vary depending on the considered temporal baselines. The 24-day temporal baseline corresponds to the maximum relative number of "active" features, while the number of "transitional" features increase with
longer baselines, more suitable to capture very slow movements.

Although our activity classification is not explicitly related to quantified displacement rates, these can be bracketed considering the corresponding temporal baselines, as suggested by several authors (Colesanti and Wasowski, 2006; Manconi, 2021; RGIK, 2020). Velocities reported in Table 5 correspond to the maximum unambiguous velocities that can be inferred for each Bt considering C band SAR measurement, with respect to ambiguity thresholds of λ/4- λ/2, respectively.


**Table 5:** number of rock glaciers belonging to each activity class and relative maximum detectable velocity.

| Landform activity class (this work) | Geomorphological activity (Scotti et al 2013) | Bt 12 days | Bt 24 days | Bt 90 days | Bt June-October |
|---|---|---|---|---|---|
| **Active** | Intact (permafrost) | 128 | 145 | 108 | 88 |
| **Transitional** | Intact (permafrost) | 162 | 145 | 182 | 202 |
| **Relict** | Relict (no permafrost) | 163 | 162 | 168 | 162 |
| **Moving debris** | - | 61 | 62 | 56 | 62 |
| **max unambiguous velocity** for λ/4 and λ/2 (cm/yr) | | 42-85 | 21-42 | 5.6-11.2 | 4.2-8.4 |



Deeper insights in the styles of activity of "active" landforms come from the examination of the number of temporal baselines at which features are active (Fig. 8). Landforms active at one specific Bt can be inferred to move at average velocity falling in narrow ranges, with maximum values captured by DInSAR depending on the considered Bt (Table 5). Periglacial features active at 12 or 24 days are characterized by typically observed displacement rates of decimetres/year (Haeberli et al, 2006),
whose DInSAR signal is lost over longer temporal baselines due to decorrelation effects. Features active only at long temporal baselines may testify slow or seasonal movements in unfavourable topographic conditions.

At the same time, activity detected over an increasing number of temporal baselines (Fig. 8) may characterize landforms undergoing seasonally variable displacement rates. Features active over very different Bt (e.g. 12 days and Jun-Oct) may indicate segmented deformation mechanisms, or an interplay of multiple environmental drivers. According to this
interpretation, landforms active over 3 or all the considered temporal baselines are characterized by the maximum spatial-temporal heterogeneity. Although a precise assessment of the nature of this heterogeneity cannot be achieved by our regional analysis, our results provide useful hints for the selection of individual cases that deserve targeted, site-specific investigations.



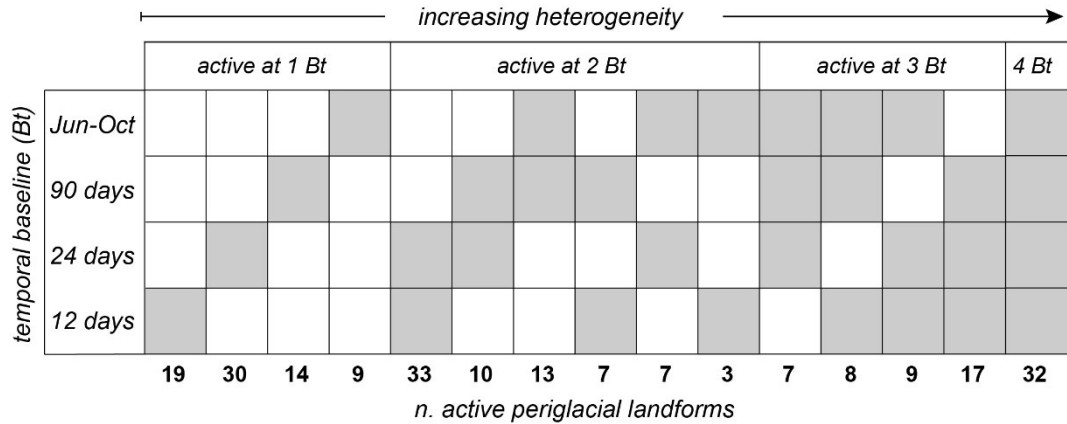

**Figure 8.** Periglacial landforms active on different temporal baselines (classification with AI_T: 0.4; see text for explanation). From left to right, landforms are active on an increasing number of temporal baselines, suggesting increasingly spatial and/or temporal complexity.

The almost constant number of "moving debris" features over different temporal baselines (Table 5) suggests that their
dynamics is not directly related to permafrost but may be simply driven by the frictional instability of slope debris, with variable displacement rates controlled by slope steepness. This class can include all those landforms, made of loose rocks and granular soils subjected to different natural processes in periglacial environments (e.g. freeze-thaw, wetting and drying), that display InSAR active deformation signals, but whose dynamics are unrelated to the permafrost of presence.

We compared our classification results at Bt=24 days (i.e. the temporal baseline that maximizes the active class) with the
geomorphological activity classification of Scotti et al. (2013). The latter only includes two classes of landforms, namely: "intact" and "relict", according to the inferred permafrost occurrence. For the sake of comparison, we grouped again our "active" and "transitional" classes in the term "intact" (Table 6). The comparison outlines that the two classifications are consistent for 458 cases out of 514. 89% of the features classified as "intact" by Scotti et al (2013) have been confirmed by our classification, while the remaining are now classified as "moving debris", based on the lack of permafrost. On the other
hand, 11% of features previously classified by Scotti et al. (2013) as "relict" has been now recognized as moving debris with contribution of InSAR and permafrost information.

Landforms classified as "active" by both the approaches usually display sharp morphological evidence with well-developed longitudinal and transversal ridges and furrows, steep frontal slopes, lack of vegetation and fresh debris cover (Fig.9a). The activity is reflected in clear DInSAR signals which, depending on the analysed Bt, can be characterized by smooth changes
with (partly) fringed patterns (Fig.9b) or decorrelated signals with a noisy pattern (Bertone et al., 2022). On the opposite, relict rock glaciers don't display any swollen convex appearance or active morpho-structures, but are less steep, frequently vegetated with shrubs and longitudinal scree accumulation testifying the presence of relict furrows (Fig. 9c). For these cases the DInSAR signal is homogenous and similar to the surrounding slope (no movement; Fig. 9d).



However, the transition from "intact" to "relict" based on DInSAR activity information can be related to an unfavourable slope

orientation to the satellite, with possible activity of N- or S-directed movements not captured by DinSAR, despite the clear

geomorphological evidence of permafrost presence (Fig. 9e,f).

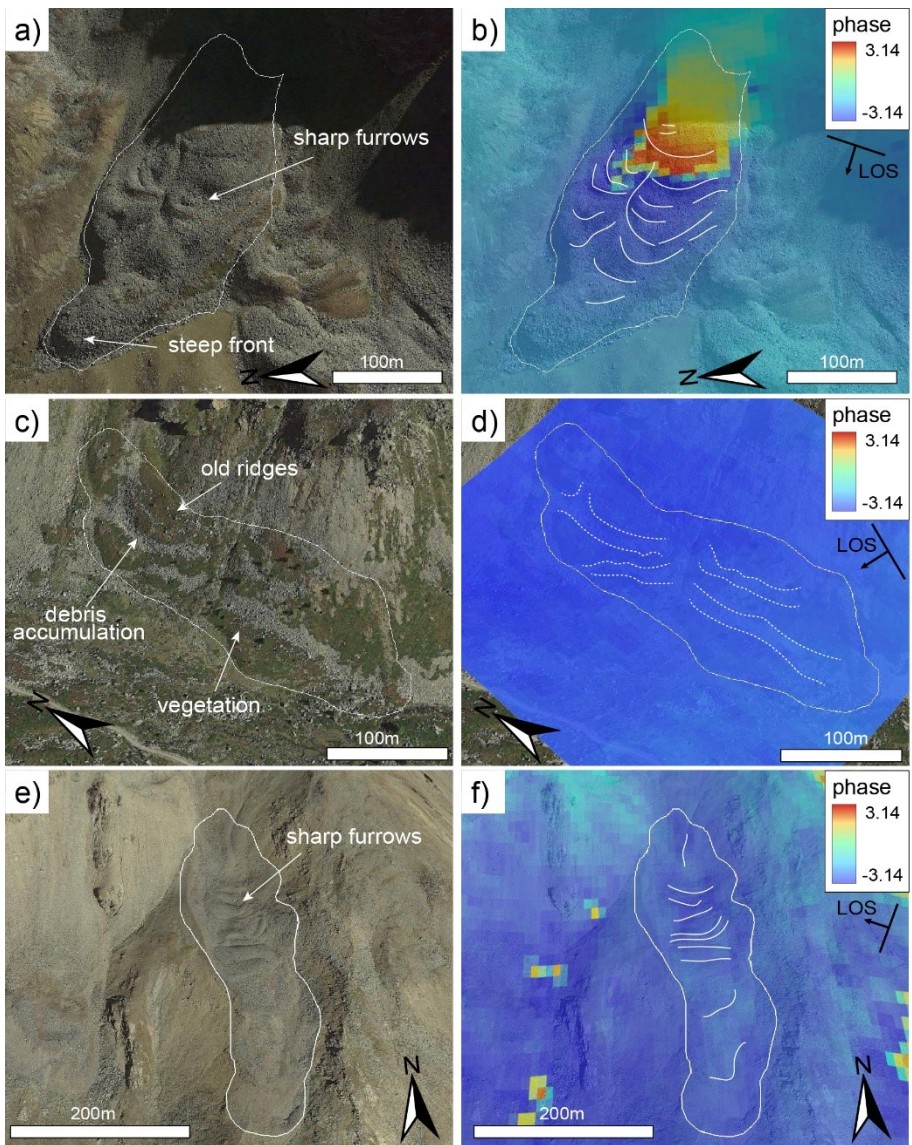

**Figure 9**: comparison between geomorphological and DInSAR-assisted activity assessment. a) active rock glacier (Bt=24days, AI=0.87),

with sharp furrows and ridges testifying by clear fringe patterns (b); c) relict rock glacier (Bt=120days, AI=0.19) characterized by abandoned and vegetated morphological evidence, mirrored in d) by the absence of DInSAR signal; e) active rock glacier with sharp furrows and lobes, facing for which the satellite is blind (C index = 0.2) resulting in f) a strong underestimation of activity underestimation of its movement (Bt=24Bt, AI=0.23). Portrayed DInSAR signals are from individual interferograms (not stacked) without APS corrections. Imagery: Google, ©2021 Maxar Technologies.





## 3.3 Distribution and controls on periglacial landform activity

Our results highlight the possibility to discriminate between periglacial landforms active at different rates, over different temporal baselines and with different complexity. Then, we analysed their distribution with respect to morphometric and land surface temperature (LST) variables, to explore possible relations with the environmental factors that control their dynamics. Periglacial landforms falling in different activity classes are characterized by topographic aspect and slope distributed in different ways (Fig. 10). Active landforms cluster along N-facing slopes, the most favourable to permafrost occurrence, although they are found also in E- and W- facing slopes when other factors (e.g. elevation, local relief, and debris supply) are suitable. Transitional landforms, still hosting permafrost but not showing significant movements, also tend to be N-facing, but show a more scattered occurrence in an aspect range between N240 and N120. On opposite, relict features are spread over broader aspect ranges (N70-N270), on S-facing slopes abandoned by permafrost. Finally, the occurrence of "moving debris" landforms doesn't show a clear correlation to aspect and is observed over a broad range of slope values. This supports the fact that they are not directly linked to the presence of permafrost and the movement can be restricted to superficial movement of rocks not necessarily related to a deeper process.

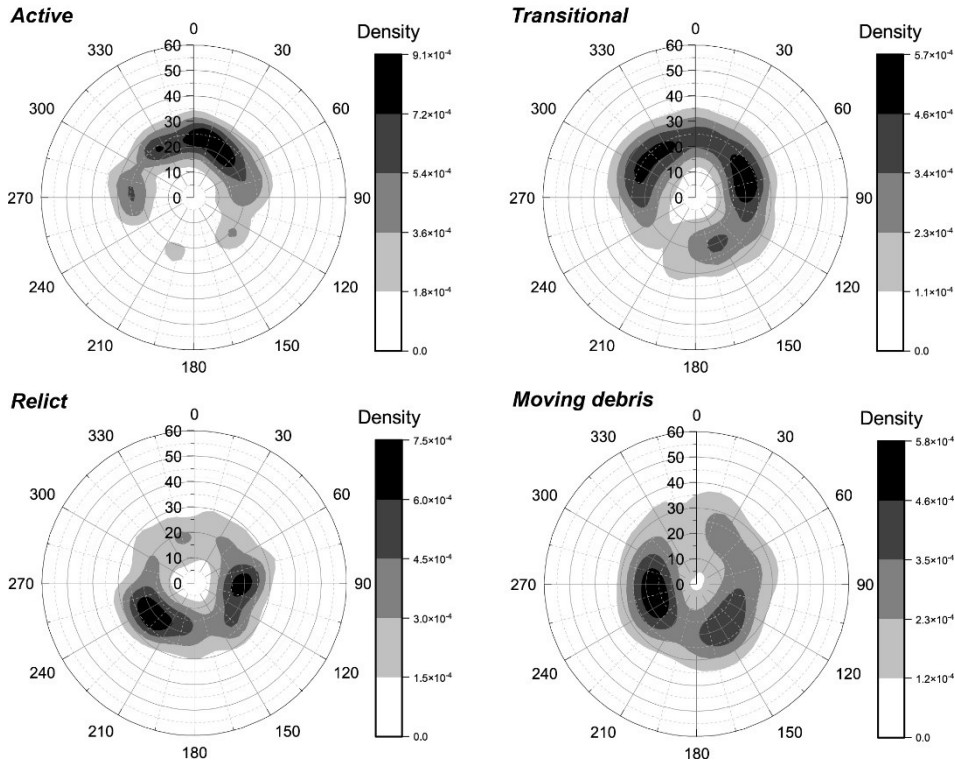

**Figure 10.** Stereographic (polar) diagrams showing the distribution of the average topographic aspect (i.e. azimuth) and slope (i.e. dip) computed for each mapped periglacial features, falling in different activity classes according to our methodology. The relative abundance of periglacial features is expressed in terms of point density.



Principal Component Analysis (Fig.11) allowed to further explore possible relationships between the state of activity of classified landforms and other morphological variables, namely: surface area, mean elevation (i.e. the elevation of the landform

polygon centroid), and the length-to-width ratio (L/W) which describes the elongation of each feature. We also considered the mean land surface temperature (LST) computed for each polygon from Landsat 8 images for the summer months (June-October), in the winter periods (October-June) and over the total analysed period (2013-2020).

We selected the first 2 principal components (PCs), accounting for 57% of the entire multivariate space, and represented the cases according to their activity class attribution in the four considered temporal baselines (Fig. 11).


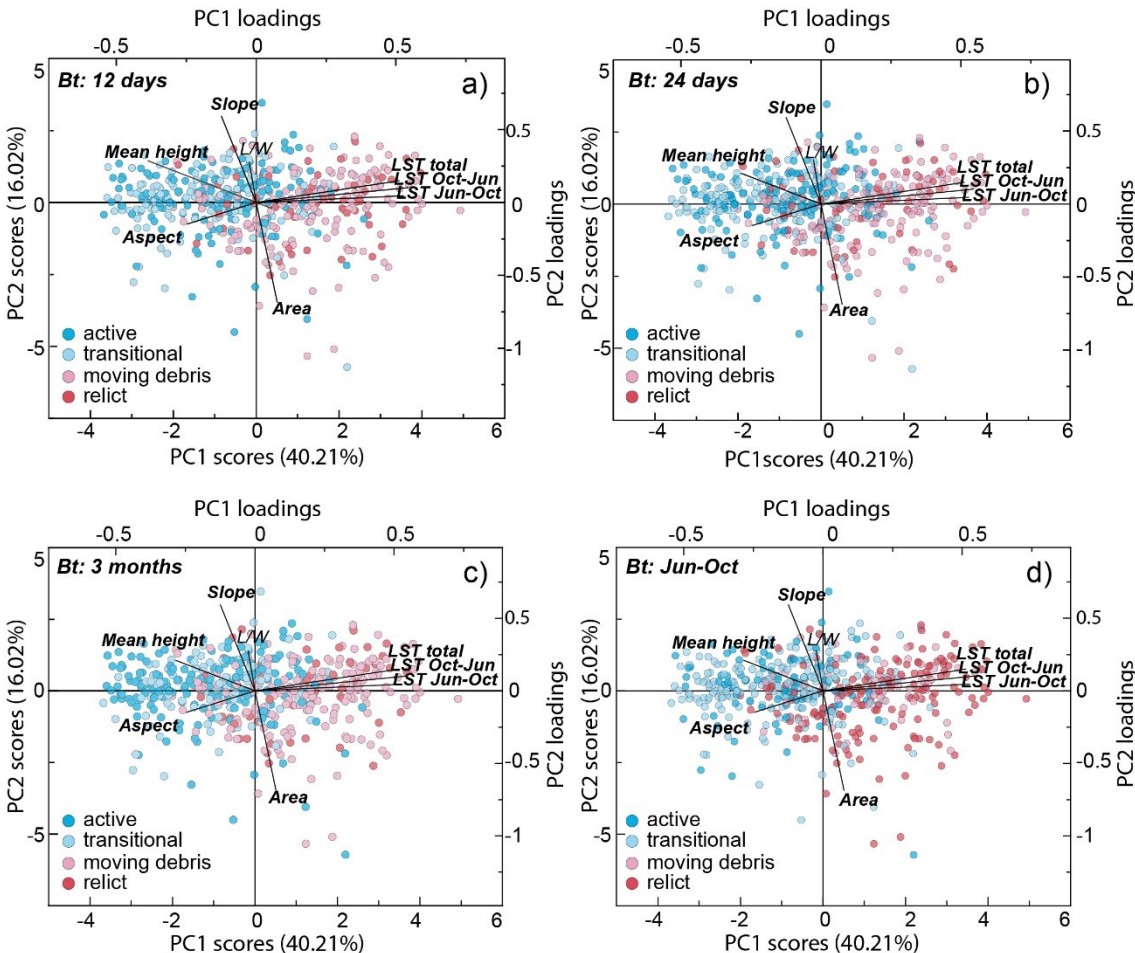

**Figure 11.** PCA biplots. The graphs portray the distribution of rock glacier activity classes with respect to the considered morphometric and LST values at a) 12 day Bt, b) 24 day Bt, c) 3 month Bt, d) seasonal Bt. The biplots show an association between activity and LST, with active features clearly trending towards low temperature.






The first principal component (PC1), accounting for about 40% of the variance, is mainly associated with the LST, with increasing temperature corresponding with less active landforms and a temperature difference of about 5°C between relict landforms, on the right size of biplots (Fig. 11). This supports the role of permafrost in the activity of the features classified as active rock glaciers and protalus ramparts, while "moving debris" features show a more scattered distribution, again supporting

that the activity of these features is less dependent on permafrost. Activity trends are also associated with mean elevation and aspect, with active and transitional landforms associated to higher elevations, and seem less dependent on slope and landform, that align with the second principal component (PC2), accounting for only 16% of the variance. The described patterns were consistent across the different temporal baselines considered in the study.

These results help outlining the typical conditions associated with each landform class, in agreement with previous findings

(Alcott, 2020; Harris and Pedersen, 1998; Mühll et al., 2003), and support the robustness of our classification approach and its ability to recognize active features whose activity may be not driven by permafrost-related processes.

## 4 Discussion

Over the last two decades, there has been growing interest in studying the state of periglacial landforms (especially rock glaciers) as important proxies of climate change, with potential implications for water availability and geohazards in alpine

environments (Marcer et al., 2021; Amschwand et al., 2020; Schoenich et al., 2015). These studies rely on the assessment of activity of large populations of rock glaciers and protalus ramparts spread over large areas. This remains challenging, and current applications of remote sensing, while expanding the investigation potential, are still hooked to expert interpretation resulting in time-consuming and potentially biased assessment (Brardinoni et al., 2019; Kofler et al., 2020).

Our study proposed an innovative, semi-automated and replicable workflow, able to classify a dataset of 514 periglacial

features, previously mapped by Scotti et al. (2013), into four classes of activity. Our classification stems from integrating two key factors: an estimation of landform activity based on the distribution of wrapped DInSAR phase difference across multiple temporal baselines, and the likelihood of permafrost occurrence as portrayed by the APIM dataset. These factors are combined into an ordinal Activity Index, that is eventually classified using a threshold value that maximizes the differences between temporal baselines.

Due to the regional scale of the study and the harsh environmental conditions of alpine settings (i.e. steep mountainous terrains, atmospheric artefacts, snow cover effects), we focused on assessing landform activity without resort to unwrapping (Delaloye et al., 2020, RGIK, 2021). In fact, identifying a stable area close enough to each periglacial feature, which can grant a reliable reference point for unwrapping algorithms, is usually very difficult in regional-scale analyses. Since unwrapped displacements are always relative to a stable point, the selection of a unique one for the entire study area would cause the propagation of

unacceptable uncertainty, amplified by topographic conditions, moving away from this point. Furthermore, alpine mass movements like rock glaciers usually display noisy DInSAR signals with poorly expressed interferometric fringes, for example





due to displacement gradients between surrounding pixels higher than half of the radar wavelength (Bertone et al., 2022), hampering the application of minimum-cost unwrapping algorithms (Carballo and Fieguth, 2000; Costantini, 1998).

Our classification approach has significant advantages and some limitations with respect to previously proposed ones (e.g.
RGIK, 2020, 2023). Our workflow is semi-automated and allows a rapid classification of large populations of periglacial landforms over wide areas (514 spread over 1000 km$^2$ in our case study), while existing approaches rely on a manual attribution of the kinematic attribute (Barboux et al., 2014; Bertone et al., 2022). Our methodology directly considers permafrost occurrence and provide results consistent with field geomorphological evidence (i.e. sharp or subdued morphology, presence of clear ridges and furrows, vegetation) and related activity classifications (Scotti et al. 2013). It allows recognizing landforms
whose activity is not directly related to permafrost dynamics (i.e. our "moving debris" class) and discriminating among landforms characterized by increasingly heterogeneous styles of activity (Fig. 8). Compared to activity assessment approaches based on multi-temporal interferometric methods (Zhang et al., 2021; Buchelt et al., 2023), our approach fully exploits the spatially distributed information contained in wrapped interferograms, avoids the uncertainties related to unwrapping, and better captures the timescales of process occurrence working on specific temporal baselines. On the other hand, the semi-
automated character of our method may suffer from local errors, which in any case are checked during the validation stage. Obviously, it also suffers the intrinsic limitations of InSAR, including the inability of SAR sensors to detect deformation signals over north and south-facing slopes (Fig. S4), hampering the detection of active landforms. Moreover, our signal enhancement approach relies on stacking, because of the difficult correction of atmospheric phase screening effects in high-mountain alpine environments.

The definition of our Activity Index is based on the frequency distribution of wrapped phase difference values emerging from background levels recorded in the adjacent stable rims. Thus, it combines information on both the extent of active areas within the mapped landforms and their unambiguous displacement rates. In this respect, the definition of "transitional" features using our methodology appears consistent with the original kinematics-based definition (RGIK, 2020, 2022; Lambiel et al., 2023), with low displacement rates resulting in Activity Index values below the detection threshold in permafrost-bearing landforms.

As a result of the proposed classification scheme, we also introduced a novel category termed "moving debris", that highlights movements that cannot be attributed to permafrost dynamics but can be interpreted as driven by the frictional instability of slope debris with variable displacement rates controlled by slope steepness.

An activity assessment methodology accounting for the information provided by InSAR over a wide range of temporal baselines (12 to 120 days) is key to gain insights into the complex spatial and temporal dynamics of periglacial landforms.
These can be either strongly heterogeneous, due to interplaying deformation mechanism (e.g. active layer movement, permafrost creep, basal sliding), or overlapping processes (Fig. 9a,b). Our regional scale results allow highlighting different styles of activity. Temporal baselines of 12 and 24 days capture the activity of rock glaciers and protalus ramparts characterized by typically observed displacement rates of decimetres/year (Haeberli et al, 2006) and up to 85 cm/yr, with the 24-day Bt corresponding to the maximum number of features classified as "active" (Table 5). Landform characterized by homogeneous
slow movements (max. 11 cm/year) are capture at longer temporal baselines (90 to 120 days). On the other hand, the activity



of a large subset of the inventory is captured in interferograms generated at different temporal baselines (Fig. 8), suggesting that they may undergo more complex seasonal mechanisms. Features classifies as active over very different temporal baselines (e.g. 12 and 120 days) may suggested segmented deformation mechanisms or multiple interplaying controls. Our results are unable to provide further insights in the actual physical processes underlying the observed styles of activity (e.g. temperature effects, hydrological forcing, destabilization), but are meant as a regional screening tool supporting the identification of case studies for site-specific investigations.

Morphometric and multivariate statistical analysis results support the robustness of our classification approach and the effectiveness of including an indicator of permafrost distribution (i.e. API map) directly in the activity classification procedure. The clustering of permafrost-bearing (active and transitional) and permafrost-lacking landforms (relict and moving debris) into distinct groups is influenced by aspect, elevation, and land surface temperature in a way consistent with the control of permafrost occurrence. Additional factors like precipitation, snow cover depth and duration could enhance the characterization of features falling in different activity classes, but this is beyond the scope of this work.

Finally, and in a more general perspective, our results prove that significant information about the activity of alpine mass movement can be automatically extracted from DInSAR wrapped phase signals, i.e. the most basic (filtered) product of InSAR processing chains, avoiding the uncertainties introduced by unwrapping procedures and atmospheric correction models not specifically designed for the alpine environment. This can open interesting perspectives towards the development of fast and automated mass movement detection tools based on increasingly powerful artificial intelligence techniques.

## 5 Conclusions

Using wrapped DInSAR phase difference and permafrost occurrence data, we provided a rapid, semi-automatic and replicable regional screening methodology, allowing to classify the activity of rock glaciers and protalus ramparts across multiple temporal baselines. This classification yielded four distinct categories: active, transitional, relict, and moving debris classes, consistent with the most recent rock glacier inventorying criteria, yet objectively and rapidly assessed, and enabled the identification of landforms exhibiting different velocity ranges and degrees of heterogeneity. The possibility to obtain a rapid classification of large populations of periglacial landforms over large areas, that is respectful of field geomorphological evidence, of kinematic and environmental constraints, provides an important tool to: a) rapidly update periglacial landform inventories; b) track the increasingly fast evolution of the alpine cryosphere; c) gain preliminary insights in the mechanisms underlying periglacial landform movements and their heterogeneity. Our approach not only produces regional maps but also serves as a valuable tool to highlight complex or fast-moving landforms in which prioritize detailed local scale analyses.

**Declaration of competing interest**

The authors declare that they have no known competing financial interests or personal relationships that could have appeared to influence the work reported in this paper.

**Acknowledgements**

We thank Alessandro Mondini for the stimulating discussions on the quantitative analysis of InSAR wrapped phase difference data. The research was partially supported by the MIUR projects "Dipartimenti di Eccellenza 2023–2027, Department of Earth
and Environmental Sciences, University of Milano-Bicocca" and "MIRAGE: Mass movement Investigation and prediction through geomorphology, Remote sensing and Artificial intelligence; PRIN22 - project number: 2022X539XM – Funded by European Union - Next Generation EU".

**Authors' contribution**

FA and CC conceived the research and designed the methodology. CC and DC prepared the geomorphological, permafrost
and LDT data for analysis and performed DInSAR processing. FA, CC and FF prepared the wrapped phase analysis workflow. FA and CC analysed and validated the results. FA and CC wrote the paper. FA managed the project.

**Data availability**

We used published or freely available data. We used the published geomorphological inventory of Scotti et al (2013; GIS data courtesy of the authors). For DInSAR processing, we used Sentinel-1 data downloaded from the ASF Vertex portal and
processed them using the ESA SNAP Sentinel-1 toolbox v.8 (The European Space Agency). The complete list of used SAR images and generated interferograms is provided in the Supplement with public access. For permafrost data, we used the freely available Alpine Permafrost Index Map (Boeckli et al., 2012). For LST, we used free Landsat 8 data acquired through the Google Earth Engine tool by Ermida et al. (2020).

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
