# Peer review of "Rapid regional assessment of rock glacier activity based on DInSAR wrapped phase signal"

_EGUsphere, 2024_

## Author Comment (AC1)

**RAPID REGIONAL ASSESSMENT OF ROCK GLACIER ACTIVITY BASED ON DINSAR WRAPPED PHASE SIGNAL**

Federico Agliardi, Chiara Crippa, Daniele Codara, Federico Franzosi

**AUTHOR COMMENTS IN RESPONSE TO ANONYMOUS REFEREE #1   (RC1)**

Dear Referee,

We wish to thank you for your careful and constructive review of our manuscript, that we greatly appreciated. Here and in the manuscript, we have addressed all the issues raised by your review. Below you find detailed responses to your comments. We took the opportunity to do additional typo corrections and text improvements.

Best regards,
Federico Agliardi
(on behalf of all the authors)
* * *
The authors present a methodology for assessing RG activity using stacked DInSAR information in a sub-region of the Central Italian Alps. The contribution is of interest to The Cryosphere. After my reading of the paper, I would like that the authors address the following points:

**Comment (1), part 1:** The rock glacier inventory (i.e., Scotti et al., 2013), used as morphological and topological benchmark for the InSAR-based characterization of RG activity, represents the main limitation of this work. The inventory, which is based on the interpretation of 2000, 2003 and 2007 optical imagery, on one hand introduces a temporal mismatch with respect to the 2017-2020 DInSAR window of interest.

> **Reply:** Our work aims at the regional-scale screening of rock glacier activity based on wrapped DInSAR phase (i.e. information on surface deformation avoiding unwrapping errors) and APIM-like maps (i.e. likelihood of permafrost occurrence). In our methodological paper, the rock glacier inventory is considered as a dataset to test our semi-automated activity assessment procedure. The periglacial features, of which we evaluate the state of activity for the period 2017-2020, were already there in 2013 and are still there now. We compare our results to the geomorphological activity classification of Scotti et al (2013), but don't use it for calibration. In this sense, there is no "integration" between the morphological and DInSAR components of the analysis. If the aim of the study were to update the inventory for practical purposes, temporal overlap should be ensured. But this is not the case.

**Comment (1), part 2:** On the other hand, the way in which it was originally conceived, and it is presently utilized in this paper, does not form a robust geomorphological reference.

In principle, the temporal mismatch makes the integration between the morphological and DInSAR components of the inventory unreliable. Currently, Scotti et al's inventory is plainly used in its original form as a topological basis for conducting the DInSAR analysis of RG activity. Indeed, some morphological components of Scotti et al's inventory need be updated since the criteria used for the compilation and classification of the periglacial landforms are unclear or do not look up to date. Among others, a key morphological attribute for distinguishing between intact RGs and other potentially similar landforms is the occurrence of a well-defined creeping front (e.g., Cicoira et al., 2021; RGIK, 2023a). Please clarify whether (or not) Scotti et al 2013 landforms (i.e., intact RGs and intact protalus ramparts) comply with this criterion distinctive of permafrost creep and shearing at depth. Equally important is the planform structure of a RG, that is, the attribute classifying the structure of a RG depending on its inherent morphological complexity.

Please clarify if your morphological inventory distinguishes between simple RGs (i.e., formed by a single and well-defined creeping lobe) and complex RGs composed of multiple lobes (e.g., multiple generations of lobes completely (or partly) overlapping to each other, and/or adjacent coalescent lobes), and based on which criteria two adjacent lobes were merged into one single RG or kept separated.

> **Reply:** regarding the robustness of the mapping criteria of Scotti et al (2013), the authors explain them very well. These criteria are consistent with previous geomorphological literature and the RGIK (2023a), although they don't include the definition of "rock glacier units". Quoting some excerpts from by Scotti et al (2013): *"For each rock glacier we have mapped the whole landform surface from the rooting zone to the foot of the front slope (Barsch, 1996)….. In the case of multiple rock glaciers coalescing into one body, …. when the frontal lobes of two (or more) rock glaciers originating from distinct source basins join downslope, we consider the two components as separate bodies ….. Where the limits between lobes are unclear and the lobes share other morphological characteristics, we classify the whole system as a unique rock glacier……Lobes originating from the same source area developing along the same flow line are considered belonging to distinct rock glaciers only if we can clearly relate them to different pulses/cohorts…..".*
>
> Scotti et al (2013) implicitly consider the morphological complexity of rock glaciers by separating or merging features depending on their estimated dynamic interaction. In the dataset of Scotti et al (2013) there are no internal sub-units, that in turn would not be considered by our present methodology. On the other hand, RGIK kinematic attribution itself is not free from uncertainties, since a kinematic attribute is assigned to rock glacier units based on the characteristics of a dominant InSAR-based "Moving Area" that rarely coincides with, or entirely covers, a rock glacier unit (RGIK, 2023a).
>
> Regarding rock glacier geomorphological activity classification criteria, here is a comparison between those used by Scotti et al (2013) and RGIK (2023a), citing respective texts:
>
> RGIK (2023a)
> Active: *If no kinematic data is available: an active rock glacier shows geomorphological evidence of downslope movement such as a steep front (steeper than the angle of repose) and possibly lateral margins with freshly exposed material on top [32, 33].*
> Transitional: *If no kinematic data is available: a transitional rock glacier has less distinct geomorphological evidence of current downslope movement than active rock glaciers in the same regional context [35, 36, 37].*

*Relict: If no kinematic data is available: a relict rock glacier shows no geomorphological evidence of recent movement. The relict state could be indicated by subdued topography, smoothed lateral and frontal slopes/margins, and by the development of vegetation and soil cover (e.g., lichen, grass, forest) [38, 39, 40]. In arid regions, vegetation may nevertheless be lacking on relict rock glaciers due to unfavorable environmental conditions [41]. Relict rock glaciers are generally found at lower elevations than active ones.*

*Scotti et al (2013)*
*Active: have steep fronts and side slopes, mostly steeper than the angle of repose of the material. The upper surface is normally covered by boulders with a micro-relief of furrows and ridges, surface expression of decelerating viscous or plastic flow due to the presence of abundant ice (Barsch, 1996).*
*Inactive: Inactive rock glaciers do also contain ice, but are no longer mobile either due to melting of most of the upper layers within the front slope (climatically inactive), or topographic constrains, and/or lack of material supply from the surrounding landscape components (dynamically inactive), while the frozen core of the rock glacier is protected from melting by the sediment cover (Barsch, 1996; Lilleøren and Etzelmüller, 2011).*
*Relict: formerly active landforms in which ice vanished. They are characterized by collapsed structures at their surface, and their surface relief is much more subdued than for intact ones (Barsch, 1996). Normally they occur around or below the current tree line, with extensive vegetation cover and a less steep front compared to the intact ones (Seppi et al., 2005; Scapozza and Mari, 2010; Lilleøren and Etzelmüller, 2011).*

As one can see, the geomorphological criteria used by Scotti et al. for "active" and "relict" are consistent with the RGIK (including the occurrence of a steep creeping front). On the other hand, the geomorphological criteria used by both for "inactive"/ "transitional" features are less sharp, as we expect in absence of quantitative measurements provided by ground instrumentation or remote sensing.

**Comment (1), part 3:** The lack of a "structure-like" attribute holds potentially critical consequences on the geomorphological interpretation of DInSAR across the range of temporal baselines used in this study. For example, the so-called "moving debris" category inferred by the authors based on the persistent rates of deformation across the entire range of baselines considered (i.e., 12d to 120+d) may derive from the activity of complex multi-lobe structures. Unfortunately, the authors do not provide sample figures showing the morphological expression of these features, both in terms of creeping front and plan structural typology.

**Reply:** We don't think that the lack of "structure-like" attributes affects the validity of our results on the regional scale. In fact, when detecting activity within mapped external boundaries of periglacial features over different combinations of temporal scales, we infer different degrees of spatial and temporal complexity, consistent with the fact that complex rock glacier units can be active on different timescales. Nonetheless, we agree with the referee that our results don't allow an in-depth interpretation of individual rock glacier deformation processes. We modified several parts of Sections 3.1, 3.2 and 4 accordingly.

Regarding the concept of "moving debris", as we also explain later in the reply to referee's comment on Lines 344-346, we simply introduced it to account for landforms that show significant deformations without being associated to permafrost occurrence. We agree with the referee that our results don't allow process-based inference, thus we modified the results and discussion sections accordingly. We just observe that these landforms: a) are characterized by significant deformation signals without being

associated with likely permafrost occurrence (APIM); b) are active independently on the considered temporal baseline; c) their occurrence is not correlated to the aspect and slope conditions typically associated with the activity of different periglacial feature categories. In this sense, the "moving debris" category, accounting for 12% of the dataset at Bt=24days, is consistent with our regional-scale methodology and requires site-specific examination when its use is aimed at updating rock glaciers inventories constructed using RGIK.

**Comments (2)**: as a follow up, inference on the possible underlying mechanisms of RG surface deformation is solely based on stacked DInSAR information i.e., tracking the temporal baselines over which landforms exhibit deformation on stacked interferograms. The foundations on which this inference rests (i.e., Figures 7 and 8) lack geomorphological support, since no information is provided on whereabout the RG footprint deformation occurs (e.g., in the rooting zone, in a very limited portion of a RG associated with thermokarst features and/or central depressions, or at the front) and on the structure of the RG footprint (e..g., simple, single front/lobe landforms vs complex, multi-lobe ones).

> **Reply:** We agree with the referee and thank him for the comment. Throughout the manuscript, we eliminated unsupported process-based interpretations (see also previous reply).

**Comment (3)**: PCA analysis, which relies on basic topographic (elevation, slope, aspect) and plan morphometric (area and length-to-width ratio) parameters, as well as on LST – poorly supports inference made at point 2, again because there are no morphological parameters (e.g., is there a well-defined front? Is there a creeping front at all? Is the RG structure simple or multi-lobe? Does the DInSAR-based activity relate to one of the lobes or to the entire landform?) involved in the analysis. Overall, the foregoing issues raised at points (1) and (2) are related to the application of a rapid approach, as opposed to a more geomorphologically sound, and "time consuming" one. As admitted by the authors in the Discussion (section 4), when one applies a rapid regional approach, inevitably limits the inherent capability of making geomorphological, process-based inferences. In this context, PCA does not really help mitigate this limitation, considering the limited set of morphological variates considered in the analysis.

As a side note, it is unclear why the authors have decided to use LST, as opposed to mean annual atmospheric temperature (MAAT). There is a wealth of empirically based literature that employs MAAT as a proxy for inferring the likely spatial distribution of permafrost in relation to the occurrence of intact and relict RGs. I understand that MAAT is incorporated in the Alpine Pemafrost Index Map, but still, this choice makes the present study not readily comparable (strictly from the MAAT characterization standpoint) with other inventories elsewhere.

> **Reply:** Thank you for the comment. PCA analysis is not intended to support the site-specific interpretation of deformation processes underlying the activity of individual features. As we have explained in different replies and in the text, this is not the aim of our study. Instead, as we now explain better in Section 2.6, "*we used Principal Component Analysis (PCA, Hotelling 1933) to explore the relationships between some variables typically associated to alpine rock glacier activity (Table 3) and the activity of periglacial landforms assessed on different temporal scales through our methodology, in order to support its validation*".
>
> Regarding the use of LST: Land Surface Temperature derived from remote sensing products over large areas is widely used to study the occurrence and state of permafrost (some references below). In our work, we don't mean to provide a detailed characterization of permafrost, nor to make comparisons to other inventories. As explained above, we include LST in our PCA as one of the variables commonly associated with different states of rock glacier activity.

Hachem, S., Duguay, C. R., & Allard, M. (2012). Comparison of MODIS-derived land surface temperatures with ground surface and air temperature measurements in continuous permafrost terrain. The Cryosphere, 6(1), 51-69.

Luo, D., Jin, H., & Bense, V. F. (2019). Ground surface temperature and the detection of permafrost in the rugged topography on NE Qinghai-Tibet Plateau. Geoderma, 333, 57-68.

Bartsch, A., Strozzi, T., & Nitze, I. (2023). Permafrost monitoring from space. Surveys in Geophysics, 44(5), 1579-1613.

Gök, D. T., Scherler, D., and Wulf, H.: Land surface temperature trends derived from Landsat imagery in the Swiss Alps, The Cryosphere, 18, 5259–5276, https://doi.org/10.5194/tc-18-5259-2024, 2024.

**Comment (4):** It is unclear how RGs characterized by bad LOS geometry (that is, poor DInSAR information on RG activity) or decorrelated signal on the shortest temporal baselines are handled in the inventory. Please clarify the extent to which in such cases more weight was attributed to the Alpine Permafrost Index Map. How did the authors treat undefined RGs? Was their activity status based on morphological interpretation? How are kinematically undefined RGs treated in the present manuscript? Please elaborate on this in the methods and eventually acknowledge relevant limitations in the results and the discussion.

> **Reply:** The aspect-vs-LOS issue is always present in every satellite InSAR application. In this case, the evaluation of periglacial feature activity is biased at slopes where real displacements diverge from the LOS directions, due to slope aspect (i.e. north- or south-facing slopes), and cannot be completely read by the radar sensor due to the intrinsic geometrical limitations of spaceborne SAR platforms.
>
> As we work with wrapped phase signals, as the direction of rock glacier movement (considered parallel to the local slope for simplicity) departs from the LOS direction we can expect that, for a given displacement, fringe patterns dilate and flatten until they become impossible to distinguish from noise. Nonetheless, this occurs gradually, and for each periglacial feature we quantify the fraction of movement that can be recorded by the sensor as a function of topographic (slope and aspect) and satellite orbit parameters (incidence LOS angle, and orbital azimuth angle), using the C index by Notti et al. (2014; supplementary Fig. S4).
>
> At the same time, the RGIK definition of "undefined" activity is subjected to a combination of rules affected by significant uncertainty. Quoting RGIK (2023):
> *"The default category is 0/Undefined. The rock glacier unit falls into this category when:*
> *• no (reliable) kinematic information is available,*
> *• the kinematic information is derived from a single point survey which cannot be related to any MA (as defined in Section 6.2.1),*
> *• the rock glacier unit is mainly characterized by an identified MA of undefined or unreliable velocity,*
> *• the kinematic information is too heterogeneous."*
>
> In these conditions, as we have already proposed for regional scale landslide applications (e.g. Crippa et al 2021, LASL), we prefer to avoid putting a precise threshold on SAR sensitivity, but provide a dataset that can be filtered case-by-case using threshold of the C index. We believe that this approach makes the results of our activity assessment more respectful of the different sources of uncertainty and make a quantitative evaluation of the impact of visibility possible. We have now explained this better in the Discussion section.

**Comment (5), part 1:** The selection of cited literature should be improved for appropriately acknowledging prior studies on the topic. Currently, some critical statements made in the introduction and the discussion of the manuscript are not well justified. Please consider revising.

> **Reply:** We are grateful to the reviewer for his careful review and detailed suggestions regarding the references (comments below), we considered all of them in the revised manuscript.

**Comment (5), part 2:** In brief, I believe the authors are teasing out process-based inferences for which they do not have enough information, considering they are presenting a rapid methodology to be applied at the regional scale. In my view, the authors could decide to revise and thoroughly update the morphological inventory (so that it can form a reliable basis for DInSAR interpretation), hence conduct supplementary analysis that fosters greater integration between the morphological and DInSAR components. Alternatively, they could recast the main objectives towards more strictly methodological aspects and region-wide implications. In this context, the case of "moving debris" is striking, especially considering that no figure with sample landforms is presented, and neither specific analysis targeting such features is included.

> **Reply:** We agree that some of the "process-based inferences" should be supported by more site-specific evidence (beyond the scope of this paper), and modified the text accordingly. See also other replies.

**Abstract**

**Comment to Lines 13-14**: "yet their rapid and reliable application over large areas is still limited". This statement is subjective and unsupported. I will develop more on this while commenting about the introduction.

> **Reply:** We agree with the reviewer, and we changed the statement to: *"yet their application to the rapid screening of rock glacier activity over large areas is still limited"*.

**Comment to Lines 19-20**: "This is combined with regional-scale information on permafrost occurrence". The authors do not provide any regional-scale information on permafrost occurrence. The Probability Index utilizes from a modelling effort that relies, among other environmental parameters, on a set of so-called permafrost evidence, including the spatial distribution of intact and relict rock glaciers (i.e., debris-covered component) and rock surface temperatures (Boeckli et al., 2012). Consequently, a circularity issue arises.

> **Reply:** We agree with the referee on the need to clearly state what we mean with "permafrost occurrence". Thus, we slightly modified the statement to: *"This is combined with regional-scale information on the likelihood of permafrost occurrence"*. Here we do not use inventory information to constrain rock glacier activity, that depends solely on InSAR-derived information and permafrost likelihood. At the same time, according to Boeckli et al (2012), the distribution of rock glaciers (classified as "intact" and "relict" similarly to what is done by Scotti et al 2013) is not an input to the APIM index but is used for calibration/validation purposes. Therefore, although the likelihood of permafrost occurrence and the distribution of intact rock glaciers is related, we don't see any circularity issue.

**Comment to Line 22**: "validated with field geomorphological observations". I could not find field-based information in the manuscript, please clarify. Visual inspection of optical spaceborne imagery does not qualify as field based.

**Reply:** With "field geomorphological observations" we meant "field evidence", that in this regional-scale study is gathered from aerial imagery. To improve clarity, we modified the statement to: *"validated with geomorphological evidence"*.

**Comment to Line 24**: "to rapidly update periglacial landform inventories". Please revise this statement. The paper provides a means for updating one attribute of a rock glacier inventory, that is, the activity status. The paper does not update the inventory, including for example the geomorphological outline associated with RG front advance. It fully relies on a 15-year-old geomorphological inventory.

**Reply:** Our methodology is aimed at assessing the state of activity of inventoried rock glaciers; thus it can contribute to updating the activity attribute of existing inventories. However, to improve the clarity we modified the statement in: *"to rapidly update the activity attributes of periglacial landform inventories"*.

**Introduction**

**Comment:** The literature cited in the introduction is in places incomplete and sometimes out of context with respect to the relevant sentences therein. Please see specific comments below. Keystone international references, such as the work conducted over the years by Haeberli and Arenson on RG thermal and mechanical behavior, appears in the reference list but is not given appropriate relevance in critical statements of the introduction. This is a problem, given the topic of this manuscript and the number of assumptions made in the interpretation of the InSAR-based results.

Also, in several instances, only one reference per topic is provided. To improve the literature context, please try to cover the range of decades through which the relevant literature has developed, acknowledging classical work appropriately, as well as more recent studies. When citing sample references within a broader set of existing studies, please consider adding "e.g." before the list of references cited.

**Reply:** We are grateful to the reviewer for his careful review and detailed suggestions (comments below), that we considered in the revised manuscript.

**Comment:** On the recent tendency to opt for rapid automated and semi-automated methodologies over manual, "time" consuming" ones, the selection is highly debatable and depends on the objectives and quality of the desired outputs. Especially when not dealing with specific catastrophic events (i.e., typhoons or major earthquakes) that require rapid spatially distributed assessment of mass wasting and/or flooding over large areas, automated methods are going to yield cartographic and thematic outputs of lower quality compared to analogues based on visual interpretation and manual mapping (e.g., Robson et al., 2020). For example, if one wishes to perform a thorough inventorying job, this requires an initial investment in time and teamwork (see the work by Way et al., 2021, in which several operators have performed a consensus-based inventorying procedure in Labrador). Secondly, visual inspection of single landforms should be considered as an initial "investment" that pays off with time in the subsequent inventory updates that will follow through the years. Indeed, visual inspection allows gaining morphological and process-oriented insights, which for the most part would go undetected when strictly applying rapid semi-automated approaches on a static morphological inventory compiled on 2000, 2003 and 2007 aerial photos (i.e., Scotti et al., 2013).

**Reply:** We thank the reviewer for this important comment, that gives us the opportunity to dispel a potential misunderstanding. We fully agree on the fact that high-quality mapping of individual rock glacier outline, morphology, structure and evidence of activity cannot be currently replaced by

automated or semi-automated methods. We also agree with the "return on investment" of manual mapping, especially in the perspective of understanding physical processes underlying field evidence.

Our semi-automated methodology is only and specifically related to the regional-scale activity classification of existing inventories (the one we use is just an example), and is conceived to: a) quickly and fully exploit the potential of InSAR data, routinely generated over large areas at different temporal baselines; b) avoid the impact of unwrapping uncertainties on the regional-scale assessment of rock glacier displacements; c) provide a screening assessment of periglacial landform activity; d) support a quick periodic update of some activity attribute of rock glacier inventories; e) identify candidate case studies for local-scale studies, accounting for displacement monitoring, landform structure and complexity / heterogeneity). We now explain this better throughout the revised manuscript.

**Comment to Lines 28-29**: "Permafrost degradation caused by climate warming changes the rheology and stability of ice-bearing soils, affecting alpine slope dynamics, sediment transport and possible destabilization (Buchelt et al., 2023)". This is an example of inappropriate referencing, given that Buchelt et al tested the reliability of DInSAR-based time series of RG velocity. In their results, they did not address alpine slope dynamics, sediment transport and destabilization. Please consider providing a broader set of more appropriate references.

> **Reply:** Thank you for the comment, we replaced Buchelt et al. (2023) with Springman et al (2012), Marcer et al (2019) and Cicoira et al (2021).

**Comment to Line 30**: Azocar and Brenning, as well as a wealth of other studies (e.g., Jones et al., 2018a; 2018b) deal specifically with RG water storage potential, as opposed to generally speaking "permafrost grounds". Please revise.

Jones, D. B., Harrison, S., Anderson, K., and Betts, R. A.: Mountain rock glaciers contain globally significant water stores, Scientific Reports, 8, 2834, https://doi.org/10.1038/s41598-018-21244-w, 2018a.
Jones, D. B., Harrison, S., Anderson, K., Selley, H. L., Wood, J. L., and Betts, R. A.: The distribution and hydrological significance of rock glaciers in the Nepalese Himalaya, Global and Planetary Change, 160, 123–142, https://doi.org/10.1016/j.gloplacha.2017.11.005, 2018b.

> **Reply:** Done, thank you very much.

**Comment to Lines 31-32**: an example of inappropriate referencing. Scapozza et al 2014 reconstruct Holocene-to-contemporary RG velocities and do not deal with implications on fast shallow slope instabilities and slope-scale natural hazards. Please rewrite or replace relevant references with others that fit and support the actual content of your sentence.

> **Reply:** Thank you, we moved the reference to a more convenient location.

**Comment to Lines 32-35**: there are a number of seminal studies that have summarized similar statements well before 2021. In the spirit of giving fair credit to prior work and providing a more exhaustive state-of-the-art, please consider adding more references at the end of this sentence.

> **Reply:** We agree and add the following references:
> Beniston, M., Haeberli, W., Hoelzle, M., Taylor, A. (1997). On the potential use of glacier and permafrost observations for verification of climate models. Annals of Glaciology, 25, 400-406.
> Haeberli, W., & Beniston, M. (1998). Climate change and its impacts on glaciers and permafrost in the Alps. Ambio, 258-265.

Kellerer-Pirklbauer, A., Kaufmann, V. (2012). About the relationship between rock glacier velocity and climate parameters in central austria. Austrian Journal of Earth Sciences 105.2.

**Comment to Lines 44-45:** Bertone et al 2022 is not concerned with and does not provide morphological-based definitions of relict and intact rock glaciers. It is unclear why this reference was added to support this sentence. Please revise.

**Reply:** Thank you, we agree. We removed Bertone et al (2022) and added Barsch (1996) and RGIK (2022).

**Comment to Lines 49-50**: "The movement of rock glaciers and protalus ramparts is dominated by internal permafrost creep and basal frictional sliding (Scapozza et al., 2011; Cicoira et al., 2021)". Scapozza et al 2011 is a seminal study on talus slopes and the distribution of permafrost but it has nothing to do and does not deal with permafrost creep and basal frictional sliding. Please support this sentence with more appropriate studies. The same observation applies to the next sentence (ending in line 53).

**Reply:** Done, thank you very much.

**Comment to Lines 56-59**: "Although a quantitative evaluation of displacement rates is a key component of the study of creeping periglacial features, a proper in situ assessment of their state of activity remains challenging, due to their difficult site accessibility, geomorphological and dynamic complexity. These factors limit the possibility to conduct geophysical surveys, boreholes and ground-based displacement measurements, that remain confined to few case studies (Bearzot et al., 2022; Bertone et al., 2023)".

The above statements contradict the bulk of past and ongoing studies conducted on single rock glaciers. In-situ investigations integrating geophysical, direct probing, numerical dating and proximal sensing (e.g., repeat UAV photogrammetric surveys, repeat airborne LiDAR surveys, and GB-SAR acquisitions) remain the best way for making progress on "the complete characterization of the state of activity of periglacial landforms", "conducted over different temporal scales and able to reflect the contributions of the different underlying deformation mechanisms" as stated by the authors in lines 53-55. An example is provided by the set of studies conducted on the Lazaun rock glacier in South Tyrol, where internal rock glacier activity has been reconstructed over millennia and monitored from hourly to daily and annual time scales (Krainer et al., 2015; Fey a<nd Krainer, 2020; Bertone et al., 2023).

Spaceborne SAR technology on one hand holds unique advantages for the regional kinematic characterization of creeping permafrost features; on the other hand, simply cannot warrant the temporal neither the spatial resolution to gain insights on single rock glacier deformation styles, especially when it comes to fast deformation rates (e.g., Buchelt et al., 2023). As per prior comment, I think the authors are reading too much from the stacked InSAR data, without providing any independent geomorphological information to support their interpretations.

Bertone, A., Seppi, R., Callegari, M., Cuozzo, G., Dematteis, N., Krainer, K., et al. 2023. Unprecedented observation of hourly rock glacier velocity with ground-based SAR. Geophysical Research Letters, 50, e2023GL102796. https://doi.org/10.1029/2023GL102796

Fey, C., and Krainer, K. 2020. Analyses of UAV and GNSS based flow velocity variations of the rock glacier Lazaun (Ötztal Alps, South Tyrol, Italy). Geomorphology, 365, 107261. https://doi.org/10.1016/j.geomorph.2020.107261

Krainer, K., Bressan, D., Dietre, B., Haas, J. N., Hajdas, I., Lang, K., et al. 2015. A 10,300-year-old permafrost core from the active rock glacier Lazaun, southern Ötztal Alps (South Tyrol, northern Italy). Quaternary Research, 83, 324–335. https://doi.org/10.1016/j.yqres.2014.12.005

**Reply**: We agree on the fact that local-scale field investigations of individual case studies provide the best data to understand rock glacier dynamics. Our statement wanted to point out the impossibility of conducting such investigations systematically for all and every inventoried rock glacier. For better clarity, we included the suggested references and rephrased our statement as follows:

*"A proper in situ assessment of the structure, mechanisms and state of activity of creeping periglacial features relies on field mapping, geophysical surveys, boreholes and ground-based displacement measurements (Krainer et al., 2015; Fey and Krainer, 2020; Bearzot et al., 2022; Bertone et al., 2023). Nevertheless, site investigations remain confined to few case studies, due to site accessibility, geomorphological and dynamic complexity, and budget constraints, that limit the possibility to systematically characterize hundreds of thousands of inventories landforms".*

**Comment to Lines 61-63**: "and lack a quantitative assessment of associated spatial-temporal kinematic patterns (Scotti et al., 2013, Buchelt et al. 2023, Bertone et al., 2022), despite the major improvements in the standardization of geomorphological mapping and activity classification (RGIK, 2021-2022)"

It is not clear how Scotti et al, Buchelt et al, and Bertone et al can be used as references to support the same sentence. Scotti et al deal with a traditional geomorphological inventory, Buchelt et al report on the deformation dynamics of a handful of RGs based on DInSAR and close range repeat topographic surveys, and Bertone et al introduce a InSAR-based methodology for the regional quantitative kinematic characterization of RGs. Please double-check this sentence for consistency and revise accordingly.

On the lack of "a quantitative assessment of associated spatial-temporal kinematic patterns" I would like to highlight that the methodology proposed by Bertone et al 2022 is quantitative and provides a standardized approach for characterizing RG velocity over a well-defined time window (i.e., a minimum of two years is required) towards the compilation of a consistent, global RG inventory. Please inspect the methods section described therein. Indeed, regional inventories compiled using Bertone et al's methodological workflow can be updated through time, by updating the characterization of moving areas across subsequent multi-temporal (i.e., biennial, or longer) time windows. This is clearly stated in the discussion and the conclusions of the paper, as well as in the RGIK guidelines (RGIK, 2023a).

> **Reply:** We agree, the references were misplaced. We now moved Scotti et al (2013) to the previous statement and eliminated Buchelt et al (2023) and Bertone et al (2022), that are cited in the following.

**Comment:** In this respect, I encourage the authors to refer to the last version of the RGIK guidelines, in which baseline concepts (RGIK, 2021) and practical concepts (RGIK, 2022) are further developed and combined in one document (RGIK, 2023a).

RGIK, 2023a. Guidelines for inventorying rock glaciers: baseline and practical concepts (version 1.0). IPA Action Group Rock glacier inventories and kinematics, 25 pp., https://doi.org/10.51363/unifr.srr.2023.002.

> **Reply:** Done, thank you very much.

**Comment:** The methodological workflow by Bertone et al 2022 builds on the quantitative assessment of "moving areas" (MAs) within geomorphological RG footprints. MAs are subdivided into kinematic classes to account for velocity variation within the annual snow-free period (early July to early October) and uncertainty associated with the spatial resolution of the interferograms (pixel size on Sentinel-1 interferogram is 20m x 20m). Subsequently, each RG unit (i.e., the single geomorphological RG footprint) is assigned a kinematic class of deformation based on the spatial extent, the location (i.e., whereabout the rock glacier footprint) and the

kinematic class of the MAs hosted therein. MA mapping and kinematic classification relies on redundancy, that is, MAs within a given RG outline are mapped only when they appear on multiple wrapped interferograms. During the MA mapping and kinematic characterization on interferograms, an iterative integration procedure with the (traditional) morphological component of the inventory (i.e., the RG outlines drawn on optical imagery) is employed. That is, when a MA is identified outside of a RG outline, the operator/s goes back to the optical imagery and controls whether or not the MA is associated with a rock glacier or with another landform.

In the manuscript, the lack of an iterative integration between the existing geomorphological inventory (i.e., Scotti et al., 2013), which is treated as static, and the InSAR-based characterization of RGs prevents a comprehensive update of the inventory itself. That is, the updating is pursued in terms of RG activity applied on 15-year old, static RG footprints.

With respect to the possibility to update regional inventories, I would like to flag that Bertone et al (2022) has a manual version and a semi-automated version. The manual version integrates the manual mapping of MAs on selected sets of interferograms and the inspection of stacked counterparts to ease the initial identification of MAs (see last paragraph on page 2775). The semi-automated version employs stacking and moving areas are automatically extracted in three sub-regions of Norway (see first paragraph on page 2777).

Manual mapping is indeed time consuming over the first iteration, but it forms an investment for the subsequent updating, which becomes easier when using the MAs mapped on the older version of the inventory. Most importantly, the manual approach over selected sets of interferograms allows: (i) interpretation of decorrelated regions (Barboux et al., 2014), otherwise not detectable on stacked scenes; and (ii) to gain geomorphic insights as one analyses the changes in intensity and spatial extent of moving areas within and around a given RG outline across a summer season and sequential summers. With respect to the second point, a number of studies have shown that RG velocity can increase over the snow-free summer months to reach an annual peak around September-October e.g., Berger et al., 2004; Delaloye and Staub, 2016; Wirz et al., 2016; Kenner et al., 2017).

Berger, J., Krainer, K., and Mostler, W.: Dynamics of an active rock glacier (Ötztal Alps, Austria), Quaternary Res., 62, 233–242, https://doi.org/10.1016/j.yqres.2004.07.002, 2004.

Delaloye, R. and Staub, B.: Seasonal variations of rock glacier creep: Time series observations from the Western Swiss Alps, in: Proceedings of the International Conference on Book of Abstracts, Potsdam, Germany, hdl:10013/epic.49110, 20–24 June 2016.

Kenner, R., Phillips, M., Beutel, J., Hiller, M., Limpach, P., Pointner, E., and Volken, M.: Factors Controlling Velocity Variations at Short-Term, Seasonal and Multiyear Time Scales, Ritigraben Rock Glacier, Western Swiss Alps, Permafrost Periglac., 684, 675–684, https://doi.org/10.1002/ppp.1953, 2017.

Wirz, V., Gruber, S., Purves, R. S., Beutel, J., Gärtner-Roer, I., Gubler, S., and Vieli, A.: Short-term velocity variations at three rock glaciers and their relationship with meteorological conditions, Earth Surf. Dynam., 4, 103–123, https://doi.org/10.5194/esurf-4-103-2016, 2016.

Although, both the methodological approach proposed by the authors and the categorial one (i.e., moving areas and RGs are classified into discrete velocity classes) proposed by Bertone et al (2022) may be used for updating RG activity at the regional scale, they do not have the spatial neither the temporal resolution to track RG velocity over time. Please see next comment on RGV tracking and ongoing efforts towards standardized monitoring.

> **Reply:** We thank the referee for these comments, that give us another opportunity to underline the differences between our approach and the different ones that have developed in the RGIK context.
> As we have now explicitly stated in the discussion, our approach is complementary and not alternative to the existing ones.

The RGIK methodologies (either manual or semi-automated) are focused on constructing and updating inventories based on a site-specific evaluation of the geomorphological and kinematics characteristics of mapped rock glaciers, accounting for the internal spatial and temporal complexity of rock glacier dynamics. In this sense, we totally agree that manual mapping is always a good investment. At the same time, as the RGIK kinematic attribution itself is not free from uncertainties, since a kinematic attribute is assigned to rock glacier units based on the characteristics of a dominant InSAR-based "Moving Area" that rarely coincides with, or entirely covers, a rock glacier unit (RGIK, 2023a).

On the other hand, our methodology aims at the rapid regional-scale screening of rock glacier activity, to support the attribution and updating of kinematic attributes to existing rock glacier inventories. Our paper is methodological and uses an existing inventory as a static dataset to test our semi-automated procedure. At this stage, we don't work at the sub-unit detail, nor do we make inferences on the underlying dynamic processes, yet we provide useful hints to the selection of relevant case studies for site-specific analyses. Future improvements could include an integration between this approach and the established methodologies proposed by RGIK (2023a).

Regarding the semi-automated methodologies by Rouyet et al. (2021) and Bertone et al. (2022), there are some differences that reflect the different scopes underlined above. As explained in detail in the Methodology section, our stacking approach considers multiple temporal baselines separately, and is applied to Sentinel1 interferograms at the higher possible multilooked resolution (about 15 m). Our approach only uses stacked median phase values, avoiding unwrapping errors, and not velocities, to avoid using classes. Our methodology relies on the automated analysis of frequency distributions of stacked median wrapped phase, that are considered feature-wise and not pixel-wise.

We have included the references suggested by the referee in relevant parts of the introduction.

**Comment to Lines 71-76**: "All these approaches strongly improved the state of the art, allowing to effectively capture the displacement rates and styles of activity of periglacial features. However, they rely on the manual analysis of multiple DInSAR interferograms and satellite optical images (Kääb et al., 2021, Rouyet et al., 2021, Zhang et al., 2021, Jones et al., 2023) or the site-specific analysis of displacement time series. Thus, they are time consuming, partly subjective, and difficult to apply systematically to regional inventory datasets, that include hundreds or thousands of phenomena, especially if the analysis is updated regularly to track the progress of climate change or geohazards".

I would like to flag that rock glacier velocity (RGV) has recently become an international reference variable (GCOS Environmental Change Variables) (ECV) for evaluating climate change effects globally (e.g., Pellet et al., 2023; Kellerer-Pirklbauer et al., 2024; RGIK, 2023b; RGIK, 2023c). RGV is implemented by building velocity time series on reference RGs that can ensure consistent sampling through time to build robust time series and forecast kinematic scenarios. In the case of InSAR-derived time series, reference RGs are characterized by simple morphologies (i.e., not complex multi-lobe RGs) and favorable LOS geometry. This RGV monitoring approach contrasts with the "regional scale" approach needed "to track the progress of climate change and geohazard" claimed in the introduction. I am not saying that the present study has no use. However, to make it more timely in light of ongoing research efforts, please consider revising the justification/need claimed therein.

Kellerer-Pirkbauer, A., Bodin, X., Delaloye, R., Lambiel, C., Gartner-Roer, I., Bonnefoy-Demongeot, M., Carturan, L., Damm, B., Eulenstein, J., Fischer, A., Hartl, L., Ikeda, A., Kaufmann, V., Krainer, K., Matsuoka, N., Morra Di Cella, U., Noetzli, J., Seppi, R., Scapozza, C., Schoeneich, P., Stocker-Waldhube, M., Thibert, E., and Zumiani, M.:

Acceleration and interannual variability of creep rates in mountain permafrost landforms (rock glacier velocities) in the European Alps in 1995–2022. Environ. Res. Lett., 19, 034022. https://doi.org/10.1088/1748-9326/ad25a4, 2024.

Pellet, C., Bodin, X., Cusicanqui, D., Delaloye, R., Kääb, A., Kaufmann, V. Noetzli, J., Thibert, E., Vivero, S. and Kellerer-Pirklbauer, A.: Rock Glacier Velocity. In Bull. Amer. Meteor. Soc. Vol104(9), State of Climate 2022, pp 41-42, doi:10.1175/2023BAMSStateoftheClimate.1, 2023.

RGIK, 2023b. Rock Glacier Velocity as an associated parameter of ECV Permafrost: Baseline concepts (Version 3.2), IPA Action Group Rock glacier inventories and kinematics, 12 pp.,

RGIK, 2023c. Rock Glacier Velocity as associated product of ECV Permafrost: practical concepts (version 1.2), IPA Action Group Rock glacier inventories and kinematics, 17 pp.

> **Reply:** We thank the referee for this important comment. We agree with it and rephrase to: a) avoid misunderstandings on the scope of our regional scale screening work; b) give proper credit for the RGV concept and related references:
>
> *"All these approaches strongly improved the state of the art, allowing to effectively capture the displacement rates and styles of activity of periglacial features. However, they rely on the manual analysis of multiple DInSAR interferograms and satellite optical images (Kääb et al., 2021, Rouyet et al., 2021, Zhang et al., 2021, Jones et al., 2023; RGIK, 2023a) or the analysis of rock glacier velocity (RGV) derived from displacement time series available for selected reference landforms (RGIK, 2023b,c). Thus, they are time consuming, partly subjective, or difficult to apply systematically to regional inventory datasets, that include hundreds or thousands of phenomena, especially if the analysis is updated regularly (e.g., Pellet et al., 2023; Kellerer-Pirklbauer et al., 2024)."*

**Methods**

**Comment to Line 110**: Table 1 and the information therein have not been used in the paper. Please consider if it is worth retaining it in the main manuscript or not.

> **Reply:** Actually, Table 1 is commented in the following text as a description of the inventory dataset. Moreover, data in Table 1 are used in Section 3.2. to compare the results of our InSAR-based classification to the geomorphological-based one by Scotti et al (2013). We have also added a reference to Table 1 in that part of the text.

**Comment to Lines 120-121:** "The activity attributes by Scotti et al (2013) are based on purely geomorphological criteria and may be affected by operator bias (Brardinoni et al., 2019)". Please consider that the issue raised in Brardinoni et al was chiefly related to the lack of standardized mapping rules in the international scientific community (e.g., do we include the rooting zone or not? How do deal with RGs composed of multiple units/lobes?). As per prior comment, once a set of international rules is established (e.g., RGIK, 2021 and later versions), then a consensus-based approach can lead to consistent mapping outputs (e.g., Way et al., 2021). In this respect, was Scotti et al compiled by multiple operators? Which geomorphological attributes were used to unambiguously classify each landform of the inventory as a RG?

Way, R. G., Wang, Y., Bevington, A. R., Bonnaventure, P. P., Burton, J. R., Davis, E., Garibaldi, M. C., Lapalme, C. M., Tutton, R., and Wehbe, M. A. E.: Consensus-Based Rock Glacier Inventorying in the Torngat Mountains,

Northern Labrador, in: Regional Conference on Permafrost 2021 and the 19th International Conference on Cold Regions Engineering, 130–141, https://doi.org/10.1061/9780784483589.012, 2021.

> **Reply:** We thank the reviewer for this comment and agree. As explained in many points above, we use the inventory by Scotti et al (2013) as a static support to test our methodology. We don't mean to construct a new inventory, update an inventory, or use the inventory for local-scale geomorphological analyses. To avoid confusion, we have now eliminated this statement on operator biases.

**Comment to Lines 28-29:** "Although not been updated after 2013, the inventory provides a robust reference to identify the location and general characteristics of periglacial landforms in the area." As per prior comment, a morphological inventory compiled on 2000-2007 photos (including the glacier extent in 2007) is going to yield spurious results when combined (or compared) aerial with SAR scenes acquired >10 years later. An example that comes to mind is the destabilized RG located by the Cancano Lakes within the area examined in this study. I am wondering the extent to which the morphological footprint in 2000-2007 is going to relate against the 2017-2020 interferograms, considering the documented multi-metric rates of front advance (Scotti et al., 2017). Given the target/justification of the present manuscript on climate change and possible RG destabilization, I believe that this temporal mismatch between the morphological inventory and InSAR kinematic information represents a critical limitation of this paper.

Scotti, R., Crosta, G. B., Villa, A. 2017. Destabilisation of Creeping Permafrost: The Plator Rock Glacier Case Study (Central Italian Alps), Permafrost and Periglacial Processes, 28, 224–236.

> **Reply:** Thank you for your comment. As already explained above, in our methodological paper the rock glacier inventory is only considered as a reference dataset to test our semi-automated activity assessment procedure. The periglacial features of which we assess the activity for the period 2017-2020 were already there in the period 2000-2007 and are still there now. Some changes of rock glacier footprints may have taken place due to destabilization, but systematic checks showed that these are negligible with respect to the evaluation of wrapped phase value distributions within the 2013 inventory polygons. Again, we underline that our present work is NOT aimed at a detailed local-scale geomorphological analysis of individual landforms. Anyway, the cited paper by Scotti et al (2017) reported that the Cancano rock glacier showed evidence of destabilization well before 2012, and its outline is consistent with that of 2017-2020. This rock glacier is N-S oriented, thus its movements are mostly not captured by SAR.

**Comment to Lines 137-139**: "We accounted for the likely extent of permafrost in the study area using the Alpine Permafrost Index Map (APIM; Boeckli et al., 2012). This is the result of a statistical model accounting for a set of permafrost occurrence predictors, including the mean annual air temperature, the potential incoming solar radiation, and the mean annual total precipitation (Boeckli et al., 2012)." And Lines 140-145.

Please consider that the modelling by Boeckli et al conducted across the European Alps relies also on a set of permafrost evidence including the spatial distribution of intact and relict rock glaciers. Therefore, a chicken-and-egg circularity issue arises. In this context, the application of a revised, more conservative permafrost index (PI) output is unclear, considering that there is no quantitative knowledge in the existing literature on how the spatial distribution of mountain permafrost may decline within a period of 10-15 years across regions of the European Alps. This approach looks not well justified, also considering that the PI conservative version has been calibrated on an outdated (i.e., 2000-2007) morphological rock glacier inventory. A last point to consider is that the Permanet index by Boeckli et al (www.permanet.eu) was meant to be applied at coarse spatial resolution. By

applying it as a predictor down to the single RG polygons, the authors are forcing an inappropriate downscaling of the probabilistic index, which is by definition unable to capture discontinuous and sporadic mountain permafrost. In the past decade, this notion has been demonstrated in the literature by several empirically based studies. Please acknowledge this intrinsic limitation in text.

> **Reply:** Please see our reply to Referee comments at Lines 19-20. Again, we want to say that we do not use inventory information as input to the activity assessment, that depends solely on InSAR-derived information and permafrost likelihood. In Boeckli et al (2012), the distribution of rock glaciers (classified as "intact" and "relict", similarly to Scotti et al 2013) is not an input to the APIM index but is used for calibration and validation purposes. As pointed out before by the referee, APIM is not a permafrost map, but a map of permafrost likelihood obtained by a statistical model. This was tested by Mercer et al (2017), that underlines its robustness and conservative character and used by Kenner et al (2017) for validation analyses on time series up to 2017. Kellerer-Pirklbauer et al (2022) also still used APIM as a reference to target study areas for monitoring. Our InSAR data are 2017-2020 (i.e. periods of enhanced activity of mass movements in the study area and before the discontinuation of Sentinel 1-B service).

> Kenner, R., Noetzli, J., Hoelzle, M., Raetzo, H., & Phillips, M. (2019). Distinguishing ice-rich and ice-poor permafrost to map ground temperatures and ground ice occurrence in the Swiss Alps. *The Cryosphere*, *13*(7), 1925-1941.
> Marcer, M., Bodin, X., Brenning, A., Schoeneich, P., Charvet, R., & Gottardi, F. (2017). Permafrost favorability index: spatial modeling in the French Alps using a rock glacier inventory. *Frontiers in Earth Science*, *5*, 105.
> Kellerer-Pirklbauer, A., Lieb, G. K., & Kaufmann, V. (2022). Rock Glaciers in the Austrian Alps: A General Overview with a Special Focus on Dösen Rock Glacier, Hohe Tauern Range. Landscapes and Landforms of Austria, 393-406.

**Comment to Lines 196-197**: "Based on the mapping criteria used by Scotti et al. (2013), we assumed that movements related to periglacial processes are confined within polygon boundaries, while surrounding areas, lacking evidence of permafrost deformation, are considered stable.

As per prior comment, this assumption is problematic, considering that the reference inventory dates back to 2000-2007, and that no integration is foreseen between the (optical based) morphological and InSAR approaches i.e., new RGs detected on interferograms, which went undetected during the visual interpretation of aerial photos, are not incorporated in a new updated version of the inventory (for example, please see iterative procedure suggested in RGIK, 2023a and a working example in which 14 newly detected RGs were included in a coeval morphological inventory (Bertone et al., 2024)).

Bertone A, Jones N, Mair V, Scotti R, Strozzi T, Brardinoni F. 2024. A climate-driven, altitudinal transition in rock glacier dynamics detected through integration of geomorphologic mapping and synthetic aperture radar interferometry (InSAR)-based kinematic information. The Cryosphere, 18, 2335–2356, https://doi.org/10.5194/tc-18-2335-2024.

> **Reply:** We thank the reviewer for this additional comment on the relationship between the geomorphological inventory and the InSAR-based activity assessment. Please see our replies to general Comment 1 and to the comments on Lines 28-29. We can add (and now have specified in Section 2.4) that we checked and calibrated the extent of the reference "stable rims" around each periglacial feature polygon to be sure we were not including deformation signal. These careful checks are among the

reasons why our methodology is "semi-automated" and not fully automated, and we now specified this in the discussion.

**Results**

**Comment:** The results are mixing up plain description on the main findings with interpretations and inferences. Please move the interpretation of the results to section 4.

> **Reply:** Done, thank you.

**Comment to Lines 311-314:** Although our activity classification is not explicitly related to quantified displacement rates, these can be bracketed considering the corresponding temporal baselines, as suggested by several authors (Colesanti and Wasowski, 2006; Manconi, 2021; RGIK, 2020). Velocities reported in Table 5 correspond to the maximum unambiguous velocities that can be inferred for each Bt considering C band SAR measurement, with respect to ambiguity thresholds of λ/4- λ/2, respectively."
What about the minimum unambiguous velocities? They are as important as the maximum ones for establishing thresholds between velocity classes, and therefore between active, inactive, and relict RGs. In this respect, the maximum baseline considered in this work is about 120/140 days. This upper temporal limit in C-band implies missing LOS displacements of 1-to-3 cm/yr (Barboux et al., 2014; RGIK, 2023b). Considering that 6-month baselines cannot be applied due to snow cover, when wishing to capture consistently 1-3 cm/yr displacements (at the lower end of inactive/transitional domain (Barsch, 1996; RGIK, 2023a) examination of 1-year interferograms is needed (i.e., from one summer to the next) to the range of study baselines. Please acknowledge this limitation, considering that in Table 5 (and the relevant text) the RGIK kinematic classification of RGs (i.e., active, transitional, and relict) is adopted.
In this context, I would like to highlight that according to RGIK (2023a), active and transitional RGs differ not only in terms of dominant annual velocity, but also on the proportion of RG surface that actually moves i.e., in order to be active, a RG should move across the majority of its surface, otherwise it would still qualify as transitional/inactive. In the present manuscript, distinction among active and inactive RGs is solely based on velocity. I see that the authors touch upon this matter when introducing equation, but I believe the readership would benefit from a more explicit statement, and perhaps from a practical visual example. Please elaborate on this.

> **Reply:** Thank you for the comment. The "minimum unambiguous velocity" that can be detected in single DInSAR interferograms is more difficult to quantify with respect to the maximum (Colesanti and Bovenga, 2006). In fact, the minimum signal that we can recognize depends on the ability to detect fringe patterns in a wrapped interferogram generated at a given temporal baseline. This depends on interferogram resolution (SAR image, multilooking, filtering, geocoding), coherence, phase gradients etc. In favorable conditions, fringes can be recognized associated to displacements > lambda/8, allowing recognizing displacement rates in the range 1-3 cm/yr at temporal baselines of 120-140 days. On the other hand, decorrelation at temporal baselines of 1 year can hamper the identification on the same signals. The temporal baselines used in this work have been selected after careful evaluation of our SAR data.

> Anyway, as already explained above, our methodology is not a RGIK spin-off and is proposed with a different approach and different scopes. That said, we specify again that our methodology is NOT based on velocity, but on the frequency distribution of stacked wrapped phase values (pixel-wise) within individual features (object-wise). The frequency of pixels characterized by phase values outside the uncertainty range of stable rims reflect both the amount of deformation (i.e. displacement in the

considered temporal baselines) and the number of pixels characterized by given displacements, and thus the extent of affected areas.

**Comment to Lines 327-331:** "Landforms active at one specific Bt can be inferred to move at average velocity falling in narrow ranges, with maximum values captured by DInSAR depending on the considered Bt (Table 5). Periglacial features active at 12 or 24 days are characterized by typically observed displacement rates of decimetres/year (Haeberli et al, 2006), whose DInSAR signal is lost over longer temporal baselines due to decorrelation effects. Features active only at long temporal baselines may testify slow or seasonal movements in unfavourable topographic conditions."

This is an overly simplistic summary of expected RG displacement rates and relevant environmental controls. Please consider rewriting in a more exhaustive fashion, while moving this part to the discussion, since it does not represent a description of the results.

> **Reply:** We thank the reviewer for this comment. These sentences are just comments on the results, the reported orders of magnitude of displacement rates are only related to the standard notions reported in Table 5 and used by RGIK.

**Comment to Lines 333-337:** "Features active over very different Bt (e.g. 12 days and Jun-Oct) may indicate segmented deformation mechanisms, or an interplay of multiple environmental drivers. According to this interpretation, landforms active over 3 or all the considered temporal baselines are characterized by the maximum spatial-temporal heterogeneity. Although a precise assessment of the nature of this heterogeneity cannot be achieved by our regional analysis, our results provide useful hints for the selection of individual cases that deserve targeted, site-specific investigations."

I appreciate that the authors touch upon the limitations of their regional approach when it comes to deciphering awkward deformation patterns. This limitation, however, contradicts some of the statements made in the introduction (e.g., lines 75-76), according to which the proposed study would fill the present gap of "tracking the progress of climate change and geohazards" over regions. Please consider revising the relevant parts of the introduction and the discussion, where similar applications and implications are introduced and discussed.

> **Reply:** Done, thank you for the comment.

**Comment to Lines 344-346:** The almost constant number of "moving debris" features over different temporal baselines (Table 5) suggests that their dynamics is not directly related to permafrost but may be simply driven by the frictional instability of slope debris, with variable displacement rates controlled by slope steepness.

As per prior comments throughout this review, this interpretation has no morphological, neither process-based support. It discounts the lack of detailed morphological information contained in the existing geomorphologic RG inventory by Scotti et al 2013, i.e., occurrence of a creeping front, and the planimetric structure of the rock glacier (simple vs complex; single lobe vs multiple overlapping and/or coalescent lobes). In my view, this limitation reflects the little integration pursued between the existing geomorphologic inventory and the InSAR-based component of this work. I suggest that the authors acknowledge this limitation.

> **Reply:** Thank you for the comment. As we now explain better in the text, we introduce the concept of "moving debris" to account for landforms that show significant displacements without being associated to permafrost occurrence. We now avoid process-based inference and just recognize that these landforms: a) are characterized by significant deformation signals without being associated to likely permafrost occurrence; b) are active independently on the considered temporal baseline; c) their

occurrence is not correlated to the aspect and slope conditions typically associated with different periglacial features categories.

**Comment to Figure 9** and accompanying text in the main manuscript: These text quoting Figure 9 contains interpretations of the DInSAR-based activity assessment and comments on the discrepancies/agreements against morphologically-based activity status classification (conducted by Scotti et al). Please make sure that a hierarchy between the morphological approach and the InSAR-based one is explicitly stated in the methods. The risk is to incur into circular reasoning that defeats the purpose of your work.

In their visual examples, the authors base their cases of agreement/disagreement by judging from the appraisal of morphological attributes such as flow-like features (ridges and furrows), or vegetation cover. It is common knowledge that flow-like features per se cannot be regarded as reliable elements for discriminating among active and transitional RGs (RGIK, 2023b). This is because the current morphological expression of RGs can typically retain formerly active (flow-like) features, leading to possibly misleading interpretations. Please consider putting more emphasis on the characteristics of the RG front(s).

In the figure caption, it is stated that atmospheric filtering was not performed. This point is also elaborated elsewhere in the text. I have nothing against this approach, but please consider that there exist atmospheric filtering approaches conceived specifically for alpine settings.

> **Reply:** Thank you for the comment. The hierarchy and relationships between the geomorphological and InSAR-based activity assessment has been better explained in the text and in the replies to previous comments above. Regarding the reported geomorphological features, the important thing that we underlined in the figure and captions is the sharpness of ridges/furrows and, more important, the steep front. This agrees with the mapping criteria and geomorphological activity attribution of both Scotti et al (2013) and RGIK (2023a). Nonetheless, we refined the caption following the referee's advice.

**Comment to Discussion:**

I appreciate the tone and approach used in this section with respect to the limitations of the proposed "rapid" approach. In this respect, please make sure that the discussion and the introduction agree with each other. When revising the manuscript, please double-check for repetitions and circular statements.

I look forward to seeing revised versions of the discussion and the conclusions reflecting the possible changes made in the introduction, methods, and the results. Thank you for the effort spent on your work.

> **Reply:** We thank very much the reviewer for his detailed and insightful review, that allowed improving the manuscript. We considered his suggestions throughout the entire manuscript revision.

**Comment to References:**

Pellet et al is not referenced in the ms but appears in the reference list.

> **Reply:** We thank the reviewer for the careful check. We removed the reference from the reference list.

---

## Author Comment (AC2)

MANUSCRIPT: EGUSPHERE-2024-1589

**RAPID REGIONAL ASSESSMENT OF ROCK GLACIER ACTIVITY BASED ON DINSAR WRAPPED PHASE SIGNAL**

Federico Agliardi, Chiara Crippa, Daniele Codara, Federico Franzosi

**AUTHOR COMMENTS IN RESPONSE TO ANONYMOUS REFEREE #2  (RC2)**

Dear Referee,

We wish to thank you for your careful and constructive review of our manuscript, that we greatly appreciated. Here and in the manuscript, we have addressed all the issues raised by your review. Below you find detailed responses to your comments. We took the opportunity to do additional typo corrections and text improvements.

Best regards,
Federico Agliardi
(on behalf of all the authors)
* * *
**General comment:** The manuscript presents a novel and timely study that introduces a semi-automatic workflow for assessing the activity state of periglacial landforms, with a focus on rock glaciers, using wrapped interferometric phase signals (DInSAR). The proposed method includes the development of an 'Activity Index' that enables categorization of landforms based on their kinematic behavior. This methodology is applied to a pre-existing rock glacier inventory across a regional scale in the Italian Central Alps. The authors validate their findings against geomorphological field evidence and investigate potential environmental controls on rock glacier dynamics through multivariate statistical analysis.

Overall, this study contributes valuable methodological insights for improving the efficiency of rock glacier kinematic classification and supports the understanding of spatial variability in periglacial landform activity. The ability to assign kinematic attributes to large inventories using remotely sensed data, as demonstrated in this study, represents an advancement in regional-scale periglacial research. However, several key issues—particularly concerning the geomorphological interpretation and physical understanding of rock glacier dynamics—should be addressed before the manuscript is suitable for publication.

   **Reply:** We thank the referee very much for recognizing the scientific contribution of our work.

**Comment 1.** The discussion of rock glacier dynamics in the manuscript, particularly in the Introduction and Discussion sections (e.g., Lines 49 and 302), places an unusual emphasis on basal frictional sliding. This is not widely recognized as the dominant process in rock glacier movement. Instead, the internal deformation and shearing within a distinct shear horizon, typically situated at depth in the permafrost core, is understood to be the principal mechanism of movement. This key concept, supported by numerous studies, is currently missing

from the manuscript and should be incorporated to strengthen the theoretical foundation of the study. The authors are encouraged to consult and reference the following publications, which provide comprehensive insights into the mechanics of rock glacier creep and kinematics:

Hu, Y., Arenson, L. U., Barboux, C., Bodin, X., Cicoira, A., Delaloye, R., Gärtner-Roer, I., Kääb, A., Kellerer-Pirklbauer, A., Lambiel, C., Liu, L., Pellet, C., Rouyet, L., Schoeneich, P., Seier, G., and Strozzi, T.: Rock Glacier Velocity: An Essential Climate Variable Quantity for Permafrost, Rev. Geophys., 63, https://doi.org/10.1029/2024rg000847, 2025.
RGIK: Guidelines for inventorying rock glaciers: baseline and practical concepts (version 1.0), IPA Action Group Rock Glacier Inventories and Kinematics, 25 pp., https://doi.org/10.51363/unifr.srr.2023.002, 2023.
Cicoira, A., Marcer, M., Gärtner-Roer, I., Bodin, X., Arenson, L. U., and Vieli, A.: A general theory of rock glacier creep based on in-situ and remote sensing observations, Permafrost Periglac, 32, 139–153, https://doi.org/10.1002/ppp.2090, 2021.

> **Reply:** We thank the referee for this comment. We totally agree and now we have explicitly included this important concept in the Introduction and in the Discussion. We also included the useful reference to Hu et al (2025), that was missing.

**Comment 2.** The background and motivation for the study, particularly in Lines 58–76, would benefit from a more comprehensive and up-to-date literature review. At present, the cited works are somewhat limited and do not fully capture the depth of recent advancements in remote sensing applications to permafrost research or rock glacier dynamics. To strengthen the Introduction, the authors should include additional relevant studies, especially those employing interferometric techniques or working toward standardizing inventory and classification methods in periglacial environments.

> **Reply:** We agree on these comments, that have been also made by Referee 1. We have updated/added references and moved them in more relevant locations when required. See also our replies to the Referee 1 comments.

**Comment 3.** The manuscript currently references an earlier version of the guidelines for rock glacier inventory and classification. The authors are strongly encouraged to adopt and explicitly reference the most recent guidelines provided by the RGIK (2023). These guidelines offer refined criteria for classifying rock glaciers based on kinematics and morphology and should be integrated consistently throughout the manuscript, including text, figure interpretations, and the reference list (e.g., revise Line 629 accordingly).

> **Reply:** Thank you for the comment. We have updated the reference to RGIK (2023a) throughout the manuscript.

**Comment 4.** RGIK (2023) states that active and transitional rock glaciers differ not only in velocity but also in the proportion of their surface area in motion. Active rock glaciers require movement across most of their surface. In this regard, the authors could enhance their analysis by leveraging the spatial information embedded in the DInSAR wrapped phase signals.

> **Reply:** Thank you for your comment. Please see our reply to the comment to Lines 311-314 made by Referee 1 above.

---

## Author Response (AR2)

**MANUSCRIPT: EGUSPHERE-2024-1589**

**RAPID REGIONAL ASSESSMENT OF ROCK GLACIER ACTIVITY BASED ON DINSAR WRAPPED PHASE SIGNAL**

Federico Agliardi, Chiara Crippa, Daniele Codara, Federico Franzosi

**AUTHORS' RESPONSE TO EDITOR'S COMMENTS**

Dear Editor,

We thank you for your review of our manuscript, that we greatly appreciated. Below we provide a detailed explanation of how we addressed your points while undertaking the requested minor revision.

Best regards, Federico Agliardi (on behalf of all the authors)

**General comment by the editor:**

.... From your author's replies and changes in the manuscript I can state that you addressed most of the reviewer's comments and especially clarified well the scope of the paper with all the advantages and disadvantages to the alternative approaches mentioned by the reviewer. Most of my comments address unclear text passages, whereby changing the wording the meaning of the statements can be clarified.

After addressing these comments, I believe that the manuscript can be quickly accepted.

**Reply:** We thank you very much for your review. Here and in the manuscript, we have addressed all the points raised by your review. We have accepted all minor phrasing and grammar suggestions and typo corrections directly in the text. Below we provide detailed responses to the remaining points.

Comment to Line 38: Vonder Mühll and Haeberli 1990

**Reply:** We thank the editor for pointing out the recurrent mistake in citing Vonder Mühll (whom we repeatedly cited as Mühll). We fixed this issue in the text and in the reference list.

Comment to Line 61: I agree with the former reviews that this statement sounds misleading, given that there are certainly more than 100 field case studies of individual rock glaciers. In your author reply you were mentioning that you mean it would be impractical to make extensive site investigations of all rock glaciers in a regional assessment, as in this present paper. To this, I agree! But then you have to formulate it accordingly, e.g.: "As site accessibility, geomorphological and dynamic complexity, and budget constraints limit the possibility to systematically characterize hundreds of thousands of inventorised landforms, regional assessments based on detailed field studies remain limited."

**Reply:** We agree with the editor and replaced the statement "Nevertheless, site investigations remain confined to few case studies, due to site accessibility, geomorphological and dynamic complexity, and budget constraints, that limit the possibility to systematically characterize hundreds of thousands of inventories landforms." with the statement "Nevertheless, as site accessibility, geomorphological and dynamic complexity, and budget constraints limit the possibility to systematically characterize hundreds of thousands of inventoried landforms, regional assessments based on detailed field studies remain limited".

Comment to Line 71: "RGIK"

Reply: Added.

Comment to Fig.3 caption: Why 240? This should be justified/mentioned in the text

**Reply:** The reason is explained at Lines 248-249: "We used a threshold of R=240, calibrated by comparing the relative frequency of intact and relict features mapped by Scotti et al. (2013) with the modelled presence or absence of permafrost (Fig.3c)". For clarity, we rephrased the entire period at lines 145-149 as follows: "Since permafrost conditions may have changed since product publication, to obtain a conservative estimate of likely permafrost extent (i.e. consider only the API class "permafrost in nearly all conditions") we recoded the map by the red (R) band values of the APIM RGB colour code (range: 0-255) and then filtered the areas with R values below a specified threshold. We used a threshold of R=240, tuned to optimize the relative frequency of intact and relict features mapped by Scotti et al. (2013) in the resulting areas of modelled presence or lack of permafrost, respectively (Fig.3c)".

**Comment to Fig.3:** How did you classify landforms, which were partly in different classes in the APIM model (e.g. a RG extent in different RGB classes) ?

**Reply:** As explained above, we used a threshold to discriminate between areas bearing or lacking permafrost, to support the activity classification proposed in Table 4. Therefore, our landforms can only fall in two classes of permafrost occurrence ("yes" or no"; Table 4). Considering that periglacial landforms can be considered active also when only a part of the mapped outline hosts permafrost (i.e. a subunit, not distinguished here), we considered outlines partly in "yes" condition as hosting permafrost for the sake of activity classification.

**Comment to Lines 206-207:** Is this sentence true for all other studies you mentioned in the previous sentence or only for the cited one (Brencher et al. 2020)? If only for the cited one, I would propose to change the previous sentence to: "Brencher et al. (2020) used a similar approach, considering ..." and not generalising it to all "other authors"

**Reply:** Done, thank you.

**Comment to Line 227:** intact rock glaciers are by definition permafrost - would it be more accurate to write "ground ice"?

**Reply:** We agree and replaced "permafrost" with "ground ice".

Comment to Line 290: temporal baselines?

**Reply:** Yes, thank you for pointing out the missing word. Fixed.

Comment to Line 311: Activity types?

**Reply:** Like in landslide science, we use the term "style of activity" to include additional information on activity other than simply "active", "inactive" etc. In this case, we comment on the distribution of landform activity over different temporal baselines and related complexity (e.g. Fig. 8)

Comment to Line 331: activity type ? Else, it is not clear what is meant by "style" in this context

**Reply:** See previous comment and reply.

Comment to Figure 8: Explain explicitly in the caption what is meant by the white and grey boxes

**Reply:** Done, thank you.

Comment to Line 381: which "values" do you mean here?

Reply: Thank you for the comment, we replaced "slope values" with "slope inclination".

**Comment to Line 407**: is the x-axis directly showing temperature differences (in °C?) as PC1? Then the unit could be given as well. What is then the unit of the y-axis (the Activity index)?

**Reply**: No, the x-axes of diagrams in Fig.11 show the (non-dimensional) value of PC1. The comment on the 5° difference between the LST of relict vs active landforms comes from the examination of our LST data, to support the comment of PCA result.

**Comment to Line 410**: this is not surprising as LST depends also very strongly on elevation - so, these two variables are not independent!

Reply: We agree. LST depends (often in non-obvious way) on many environmental and topographic factors. PCA results here allow identifying some of these dependences and validating our classification on them, too.

**Comment to Line 435:** what is meant by that? ("ordinal index")

**Reply**: Here we mean that the Activity Index is a continuous scalar number and not a discrete categorical one. In fact, we need to specify the threshold AIT to obtain a discrete classification (active, inactive, transitional, relict). For better clarity, we replaced "ordinal" with "scalar".

Comment to Line 453: what is meant by "Compared" here? "Similar to" or "In contrast to"?

**Reply**: We thank you for outlining a possible ambiguity. We replaced "compared to" with "different from".

Comment to Line 457: See comment to previous sentence: Do you mean here "Similar to" or "In contrast to"?

**Reply**: See previous comment and reply.